# Diverse landscapes beneath Pine Island Glacier influence ice flow

Robert G. Bingham [1], David G. Vaughan[2], Edward C. King[2], Damon Davies[1], Stephen L. Cornford[3],
Andrew M. Smith [2], Robert J. Arthern[2], Alex M. Brisbourne[2], Jan De Rydt[2], Alastair G.C. Graham [4],
Matteo Spagnolo[5,6], Oliver J. Marsh[7] & David E. Shean [8]

The retreating Pine Island Glacier (PIG), West Antarctica, presently contributes ~5–10% of global sea-level rise. PIG's retreat rate has increased in recent decades with associated thinning migrating upstream into tributaries feeding the main glacier trunk. To project future change requires modelling that includes robust parameterisation of basal traction, the resistance to ice flow at the bed. However, most ice-sheet models estimate basal traction from satellite-derived surface velocity, without a priori knowledge of the key processes from which it is derived, namely friction at the ice-bed interface and form drag, and the resistance to ice flow that arises as ice deforms to negotiate bed topography. Here, we present high-resolution maps, acquired using ice-penetrating radar, of the bed topography across parts of PIG. Contrary to lower-resolution data currently used for ice-sheet models, these data show a contrasting topography across the ice-bed interface. We show that these diverse subglacial landscapes have an impact on ice flow, and present a challenge for modelling ice-sheet evolution and projecting global sea-level rise from ice-sheet loss.

[1] School of GeoSciences, University of Edinburgh, Edinburgh, EH8 9XP, UK. [2] British Antarctic Survey, Cambridge, CB3 0ET, UK. [3] Department of Geography, College of Science, Swansea University, Swansea, SA2 8PP, UK. [4] College of Life and Environmental Sciences, University of Exeter, Exeter, EX4 4RJ, UK. [5] School of Geosciences, University of Aberdeen, Aberdeen, AB24 3UF, UK. [6] Department of Earth and Planetary Science, University of California, Berkeley, CA 94720-4767, USA. [7] Gateway Antarctica, University of Canterbury, Christchurch 8140, New Zealand. [8] Applied Physics Laboratory, University of Washington, Seattle, WA 98105-6698, USA. Correspondence and requests for materials should be addressed to R.G.B. (email: r.bingham@ed.ac.uk)

Over 40 years of satellite observations, the West Antarctic Ice Sheet (WAIS) has consistently lost ice and contributed up to a 10th of the observed rise in global sea levels[1–6]. The potential for this contribution to accelerate further as a consequence of the ice sheet's dynamical instability[7, 8] poses a threat to the long-term security and prosperity of the planet's coastal populations and infrastructure[1]. However, despite advances in Earth-system modelling to constrain future change, recent projections of the Antarctic Ice Sheet's future global sea-level contribution still have a significant uncertainty[9, 10]. One of the greatest sources of this uncertainty is the parameterisation of basal boundary conditions in the ice-sheet models used for projection[11, 12]. This is most acute for the fast-flowing ice streams that discharge >90% of ice from Antarctica. A particular challenge for these models is the simulation of the basal traction generated by ice flow over its bed, which is the primary restraint on ice flow[11–15]. Few observational data exist beneath Antarctic ice streams at the sub-km scale required to parameterise this complex, but crucial, interaction.

Here, we present high-resolution images of the topography underlying West Antarctica's Pine Island Glacier (PIG; Fig. 1). This glacier has undergone sustained retreat since at least the 1940s[16], with thinning having propagated progressively upstream from the floating ice shelf, along the main trunk and into its tributaries over the interior basin[6]. Several modelling studies have projected that the retreat and progressive upstream thinning of PIG will continue in the coming decades, potentially increasing its contributions to global sea-level rise[7, 17–20]. However, the magnitude and rate of the projected contribution critically depend on the pace at which the retreat may propagate into the interior basin, which is largely controlled by the poorly constrained basal boundary conditions[12, 14, 17, 21].

Our results reveal that PIG is underlain by a diverse landscape, and that the roughness of the landscape at the short wavelengths revealed by our observations impacts directly upon the ice flow. This demonstrates the predominance of form drag on ice flow, which presents a challenge for modelling the future retreat of the ice and, in turn, projecting global sea-level rise from ice-sheet loss.

## Results

**High-resolution images of Pine Island Glacier bed.** Our data comprise the first set of radar surveys that captures sub-kilometre-scale basal topography across an Antarctic glacial catchment, with a total coverage of ~1500 km$^2$, or ~15%, of PIG's main trunk and tributaries (defined as surface ice flow >200 m a$^{-1}$). The data, obtained by 1–3 MHz over-snow radar during three austral field seasons (2007/08, 2010/11 and 2013/14), were typically acquired in 10 × 15-km patches. The data comprise a total of nine patches imaged at 40 m (cross-flow) × 100 m (along-flow) grid resolution (Fig. 1d–l) and, as a complete set, constitute by far the largest and most spatially detailed observational data set of a contemporary subglacial landscape ever acquired. Full details of the radar data acquisition and processing, and generation of these images, are given in the Methods.

The data reveal diverse subglacial environments across PIG that contrast with the smooth interpolated bed from prior airborne radar surveys (Fig. 2), and which has been used to define the boundary condition in most ice-sheet modelling of the last decade[20, 22–24]. All but one of the sites displays lineated bedforms aligned with the current ice flow, which are typical of features found on palaeo-ice-stream beds[25–30] (Fig. 2). The exception is in the inter-tributary slow-flowing area (Fig. 1k), where no streamlined topography is seen. Broadly, these observations confirm previous assertions that the pattern of PIG's ice-flow

configuration has not changed significantly for thousands of years, despite variations in ice discharge[22, 31]. Between the different patches, however, the character and dimensions of the lineations vary (Table 1). Beneath the main trunk, subglacial lineations have vertical heights of ~2–10 m (measured trough-to-crest) and typical widths of 2–400 m redolent of 'Mega-Scale Glacial Lineations' (MSGL) moulded into subglacial sediments[25–28] (Figs. 1c, d, f, g and 2b, c; Table 1). Beneath the southern tributaries, much larger lineated bedforms are present, with vertical heights of ~25–80 m and widths of 300–1000 m (Fig. 1j, l; Table 1). In some locations, the smaller lineations are superimposed over larger basal features, for instance in the main trunk where an ~5-km-wide, ~400-m-high subglacial protuberance projects upwards from an otherwise flat bed (Fig. 1e) and in the southeastern tributary t5 where ~1-km-wavelength MSGL are superimposed over, and sometimes oblique to, a set of along-flow lineations with an ~5-km spacing (Fig. 1g).

The observed heterogeneity in sub-kilometre subglacial topography likely results from a combination of glacial activity and the inherited underlying geology. Seismic surveys have revealed that soft, deformed subglacial sediments are ubiquitous beneath the ice in all of our surveyed regions[32, 33], regardless of the amplitude of the radar-imaged subglacial features. The vertically subtle MSGL present in most of our patches have dimensions consistent with MSGLs from other settings that have been interpreted as the topographic expression of ice flow over deformable beds[27, 30, 34, 35] (Fig. 2). The larger features in the subglacial landscape, such as the ~400-m-high protuberance within the central trunk (Fig. 1e), the >300-m cliff which basal ice overrides in one southern tributary (Fig. 1j) and the >100-m-high ridges that are seen within both southern tributaries (Fig. 1j, l), all have amplitudes far exceeding those of MSGL recorded from any deglaciated setting[30], and probably express components of the underlying geology, draped by deforming sediments a few 10 s of metres thick[33]. PIG's trunk is bounded to the south by gravity anomalies indicative of a thicker crust, often associated with a harder subglacial bedrock[36], and one explanation for the rougher basal topography observed in PIG's southern tributaries (Fig. 1h–j) is that the substrate here is substantially tougher to erode and the subglacial landscape is less mature.

**Implications for ice-sheet modelling projections.** The magnitude of ice flow through much of PIG dictates that the major component must result from basal motion, being a combination of sliding of ice over its bed, and deformation of the uppermost layers of that bed[37]. Under such conditions, the resistive force exerted by the bed on the overlying ice sheet involves two processes: basal friction at the ice/bed interface and/or in a layer of deforming basal sediments[38]; and form drag, the resistance to ice flow as ice deforms across and around basal obstacles[39]. A fully three-dimensional ice-sheet model might naturally simulate form drag arising from basal features longer than its horizontal resolution, but cannot be expected to simulate form drag not represented in the subglacial topographic model on which it rests nor, indeed, due to features of wavelengths shorter than it can resolve. Since neither basal friction nor form drag can currently be predicted from observable parameters, most models are initialised by an inversion which yields a field of basal traction (a.k.a. slipperiness) that allows the model to match the observed ice-surface velocity and/or elevation changes[11–15, 24, 37]. It is clear that the short-wavelength bed roughness shown in our data can explain much of the variability in the traction coefficient ($\beta$ in Fig. 1d–l; Table 1) derived by the initialisation inversion in one model[37] (Fig. 3a). Given that seismic data indicate broadly similar

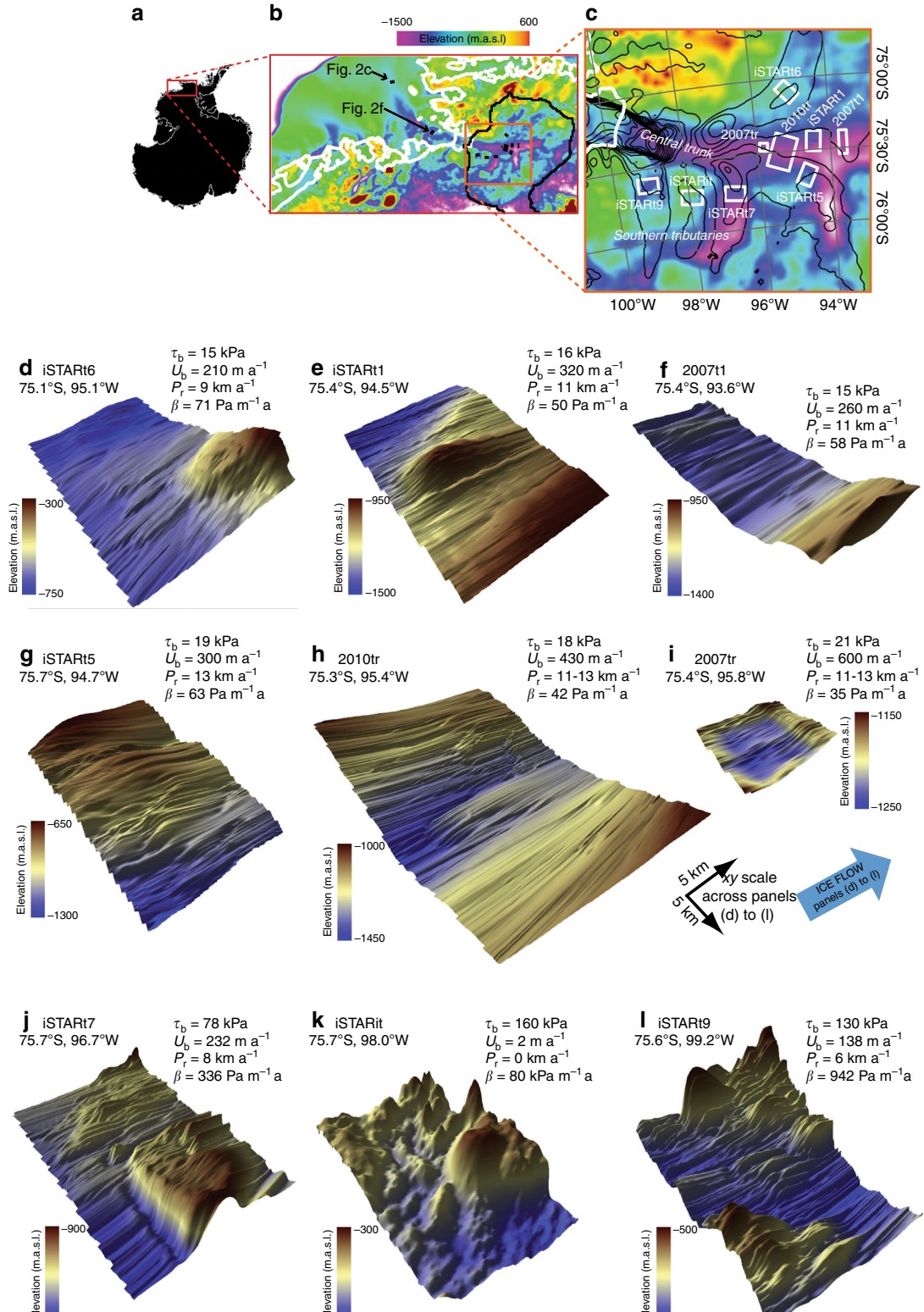

**Fig. 1** High-resolution images of the bed across Pine Island Glacier. **a** Location and context. In **b**, the colourmap shows regional bed topography from Bedmap2[23], the black line is the ice divide, the white line is the grounding line[51], and high-resolution survey patches are shown as black rectangles. Locations of offshore bathymetry shown in Fig. 2c, f are marked. **c** uses the same schema but demarcating survey patches with white rectangles, labelled by season of data acquisition (2007/08, 2010/11 and 'iSTAR' = 2013/14) and an end label denoting the location (where 'tr' = trunk; 'it' = intertributary and 't1, t5...' denotes tributaries numbered after ref. [52]. Surface ice velocities[53] contoured at 100-m intervals are also shown. **d-l** Perspective views of the bed beneath Pine Island Glacier, together with parameters of ice flow. Vertical exaggeration in all images = 10. $\tau_b$ and $U_b$ are the mean basal shear stress (kPa) and mean basal ice velocity (m a$^{-1}$) from model inversion[37]; $P_r$ is the measured upstream propagation rate of ice thinning per ice-stream tributary from 1992 to 2015 using a thinning/non-thinning threshold of 1.0 m a$^{-1}$ [6] and $\beta$ is the inverted basal traction coefficient equal to $\tau_b/U_b$

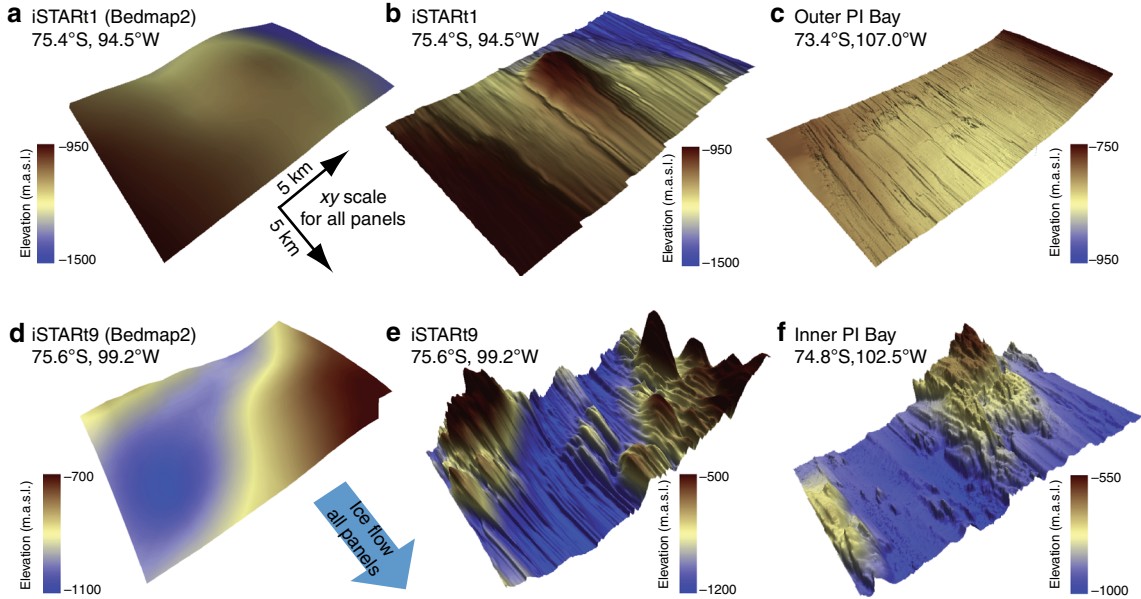

**Fig. 2** Comparison of surveyed beds beneath Pine Island Glacier with those used in ice-sheet modelling and imaged in selected palaeo-ice-streams. **a** Bed topography at site iSTARt1 (location in Fig. 1) from previous knowledge[23]. **b** New bed topography (alternative perspective view to Fig. 1e). **c** Analogous subsample of bed topography from outer Pine Island Bay imaged from data presented in ref. [27]; patch location marked on Fig. 1b. **d** Bed topography at site iSTARt9 (location in Fig. 1) from previous knowledge[23]. **e** New bed topography (alternative perspective view to Fig. 1l). **f** Analogous subsample of bed topography from inner Pine Island Bay imaged from data presented in ref. [29]; patch location marked in Fig. 1b

properties in subglacial sediments across the PIG catchment[32, 33], we conclude that much of the variability in the basal traction must be related to unresolved bed topography.

## Discussion

Many state-of-the-art Antarctic ice-sheet models use the 'Bedmap2' basal topography[23] or derivations thereof[40]. This topography was derived from unevenly distributed ice-bed elevation measurements spaced between tracks by many ice thicknesses, and interpolated onto a regular 5-km grid and, although supplied at 1-km resolution, only 17% of 5-km cells in Bedmap2 contained measurements[23]. Thus, Bedmap2 contains limited information on bed roughness, even for length scales >10 times the ice thickness. Our data thus show that significant, and potentially influential, bed topography occurs throughout PIG (and presumably other ice streams) on length scales of ~5–50% of the ice thickness.

The issue of the degree to which basal traction arises from friction or form drag becomes more significant when models are used for projection. To achieve projections, most models are run from the initialised condition with an unchanging field of traction coefficient varying according to a heuristic parameterisation of bed rheology, ranging from linear viscous to plastic. The choice of this parameterisation has a substantial impact on the timing and magnitude of ice loss[12]. However, while basal friction is likely to be a highly dynamic field, evolving as water melts, flows and refreezes, and subglacial till is mobilised and refrozen[41], form drag is likely to be considerably more static, defined primarily by the size and orientation of bedrock undulations and protuberances. Recent changes in PIG already indicate the potential importance of this factor, in as much as satellite data have shown that the rate of upstream propagation of ice thinning on PIG varies considerably between the tributaries, the southerly tributaries with rough beds showing 2–3 times slower propagation than the smoother tributaries[6] (Table 1; Fig. 3b). In summary, the

new data show that the basal traction (hence ice flow) of Pine Island Glacier is much more heavily influenced by form drag, i.e., as opposed to basal friction, than has previously been shown to be the case.

Our data provide insight into the topographic diversity that exists even within one subglacial basin, let alone the entire ice sheet, which cannot adequately be represented in ice-sheet models. Given that the basal boundary is already identified as a major source of uncertainty in model projections[17, 21], they expose an urgent need to develop techniques for efficient measurement or more intelligent indirect estimation of short-wavelength subglacial topography beneath other vulnerable ice streams. One prospect for recovering short-wavelength subglacial topography may be provided by airborne swath–radar techniques that are currently under development[42]. Until such independent evaluation of form drag and basal friction is integrated into models, the current generation of ice-sheet models will be hampered in establishing projections of ice loss and sea-level rise. As an immediate step, the new data now make it possible to run data-informed experiments to develop adequate parameterisations of short-wave form drag on large outlet glaciers and ice streams, and in doing so expand our theoretical knowledge of its effects on ice flow, building upon existing idealised treatments[43, 44].

The issues that we highlight will be particularly acute in PIG's neighbour, Thwaites Glacier, which holds the potential for rapid and irreversible retreat, and a considerable contribution to sea-level rise[8]. Thwaites Glacier's lower reaches appear to exhibit a similarly high basal traction to the roughest of our patches[11, 37], and may thus contain a similarly dramatic basal topography, but apparently already show a more rapid inland propagation of thinning than even the smoothest tributaries on PIG[6]. Here, the significance of the interplays between basal topography, which may be sufficient to pause the retreat of the grounding line, and the static and dynamic contributions to basal traction, have yet to be explored.

**Table 1 Parameters pertaining to ice flow and basal topography across each survey patch**

| | Mean basal shear stress ($\tau_b$)/kPa | Mean basal ice velocity ($U_b$)/m a$^{-1}$ | Measured upstream propagation of thinning ($P_r$)/km a$^{-1}$ | Inverted basal traction coefficient ($\beta$) | Dominant vertical height of topography/ m | Typical width of streamlined features/m | Normalised across-flow roughness using a 2-km moving window | Normalised along-flow roughness using a 2-km moving window |
|---|---|---|---|---|---|---|---|---|
| iSTARt6 | 15 | 210 | 9 | 71 | 2–6 | 450–550 | 0.08 (new)0.10 (BM2) | 0.001 (new)0.04 (BM2) |
| iSTARt1 | 16 | 320 | 11 | 50 | 2–3 | 100–300 | 0.77 (new)0.24 (BM2) | 0.72 (new)0.04 (BM2) |
| 2007t1 | 15 | 260 | 11 | 58 | 2–6 | 200–400 | 0.28 (new)0.09 (BM2) | 0.04 (new)0.002 (BM2) |
| iSTARt5 | 19 | 300 | 13 | 63 | 3–6 (MSGL) 100 (larger features) | 500–800 (MSGL) 5000 (larger features) | 0.43 (new)0.24 (BM2) | 0.07 (new)0.16 (BM2) |
| 2010tr | 18 | 430 | 12 | 42 | 5–10 | 300–400 | 0.18 (new)0.14 (BM2) | 0.07 (new)0.08 (BM2) |
| 2007tr | 21 | 600 | 12 | 35 | 2–6 | 200–300 | 0.05 (new)0.08 (BM2) | 0.04 (new)0 (BM2) |
| iSTARt7 | 78 | 232 | 8 | 336 | 5–10 (MSGL) 25–50(larger features) | 300–500 (MSGL) 700–900 (larger features) | 0.13 (new/upstream) 0.27 (new/ downstream)0.22 (BM2) | 0.02 (new)0.05 (BM2) |
| iSTARit | 160 | 2 | 0 | 80,000 | n/a | n/a | 0.59 (new)0.17 (BM2) | 0.90 (new)0.21 (BM2) |
| iSTARt9 | 130 | 138 | 6 | 942 | 5–12 (MSGL) 50–80 (larger features) | 300–600 (MSGL) 600–1000 (larger features) | 1 (new) 0.22 (BM2) | 0.67 (new)0.12 (BM2) |

*FFT* fast Fourier transform

$\tau_b$ and $U_b$ are the mean basal shear stress (kPa) and mean basal ice velocity (m a$^{-1}$) from model inversion[37]; $P_r$ is the measured upstream propagation rate of ice thinning per ice-stream tributary from 1992 to 2015 using a thinning/non-thinning threshold of 1.0 m a$^{-1}$ [6] and $\beta$ is the inverted basal traction coefficient equal to $\tau_b/U_b$. Columns 6 and 7, respectively, list the dominant vertical heights (measured trough-to-crest) and typical widths of streamlined features; in the cases of patches iSTARt5, iSTARt7 and iSTARt9, two sets are listed, where the smaller values pertain to likely MSGL and the larger values pertain to the longer-wavelength lineations over which the MSGL are superimposed. Most lineations continue beyond the patch boundaries rendering us unable to provide typical lengths. Columns 8 and 9 list normalised FFT-derived roughness values retrieved from profiles taken across and along flow, respectively, using 2-km moving windows—see Methods summary for more details. For iSTARt7, two across-flow profiles were measured, one across the relatively flat upstream region and one across the higher elevation, rougher downstream portion

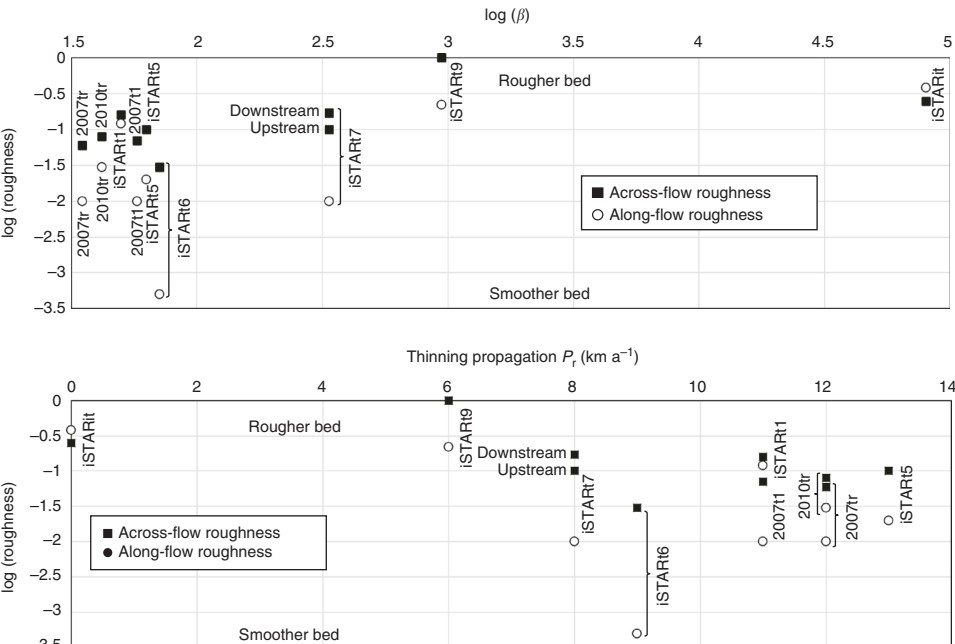

**Fig. 3** Influence of bed roughness on inverted basal traction and inland propagation of ice-stream thinning. Table 1 caption describes the derivation of basal traction $\beta$ and inland propagation of thinning $P_r$. Basal roughness has been measured across and along flow at each patch in 2-km moving windows. In all cases except site iSTARit, the along-flow profiles are smoother than across-flow profiles, as expected in locations of streamlining

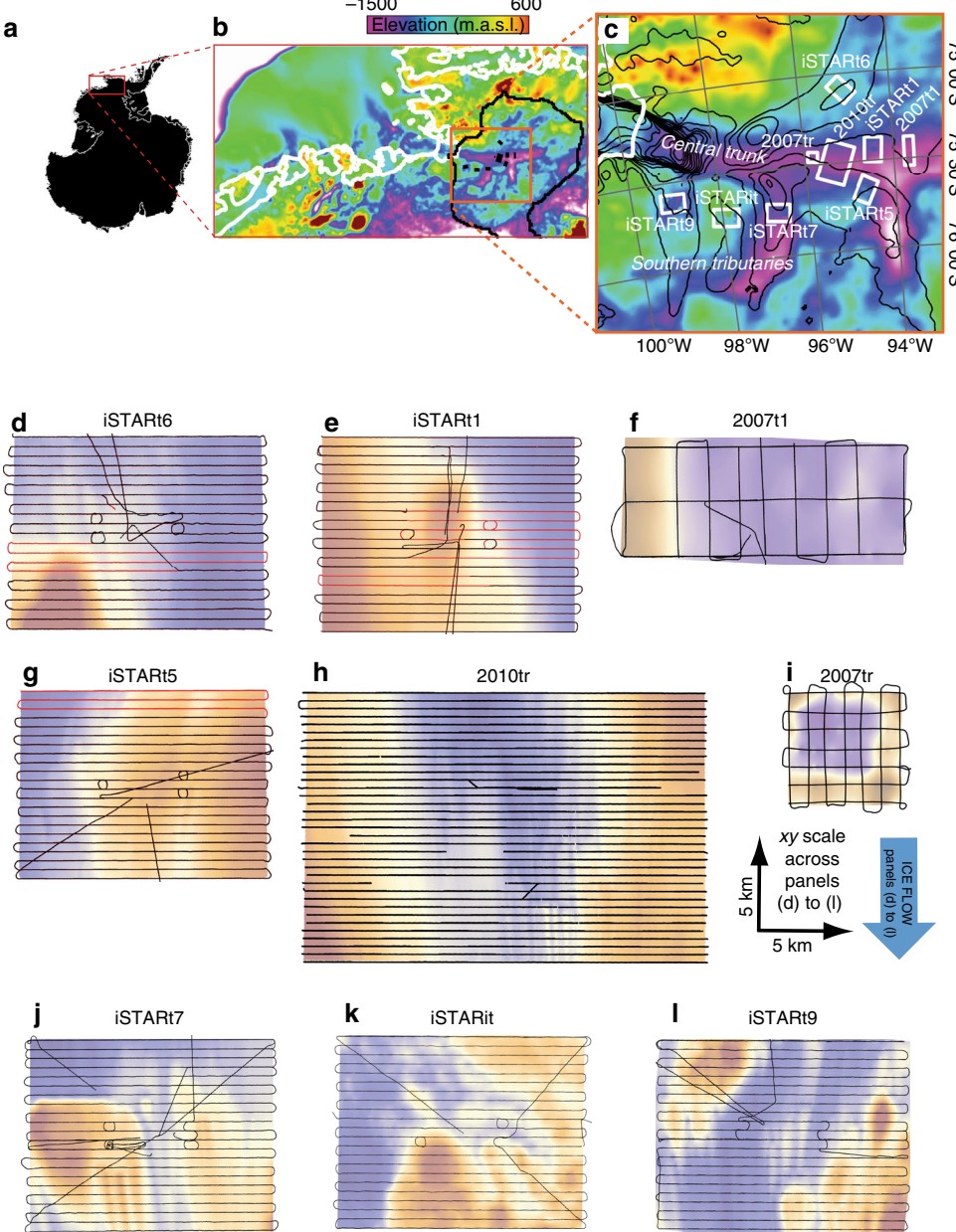

**Fig. 4** Radar coverage. **a** Location and context. In **b**, the colourmap shows regional bed topography from Bedmap2[23], the black line is the ice divide, the white line is the grounding line[51] and high-resolution survey patches are shown as black rectangles. Locations of offshore bathymetry shown in Fig. 2c, f are marked. **c** uses the same schema but demarcating survey patches with white rectangles, labelled by season of data acquisition (2007/08, 2010/11 and 'iSTAR' = 2013/14) and an end label denoting the location (where 'tr' = trunk; 'it' = intertributary and 't1, t5...' denote tributaries numbered after ref. [52]). Also shown are surface ice velocities[53] contoured at 100-m intervals. **d–l** Plan views of each radar-survey patch across PIG. The colourmap shows bed elevation (scales in Fig. 1). Black lines depict radar tracks with dual-frequency GPS navigation. Red lines in **d**, **e**, **g** depict traverses where the dual-frequency GPS failed and navigation was recovered as described in the Methods summary

## Methods

**Radar data acquisition**. All the radar data were collected with British Antarctic Survey DEep-LOoking Radio Echo Sounder (DELORES) monopulse radars towed by snowmobiles operated at a constant speed of $12 \pm 2\,\mathrm{km\,h^{-1}}$. Each system has a transmitter that fires a $\pm2500$-V pulse into resistively loaded dipole antennae at a firing rate between 1 and 5 kHz, and a receiver that registers the returned signals at identical dipole antennae and records them on a ruggedised computer equipped with oscilloscope software. A dual-frequency GPS was used to record the $x$, $y$, $z$ coordinates of the centre point between the transmitter and receiver; from 2010, an additional single-frequency GPS was integrated into the system to collect duplicate $x$, $y$, $z$ information primarily to facilitate rapid field quality control checks and also to store back-up navigation in the event of dual-frequency GPS failure.

Data from 2007/08 were acquired with 40-m half-dipole antennae, giving a centre frequency of ~1 MHz, and the transmitter was fired at a pulse-repetition rate of 3 kHz. Data from 2010/11 and 2013/14 were acquired with 20-m half-dipole antennae, giving a centre frequency of ~3 MHz, with the transmitter fired at 1-kHz repetition rate. In all years, however, multiple radar returns were first stacked in the oscilloscope buffer to reduce the signal-to-noise ratio, so that each trace of the final record was built up of the data acquired over an along-track distance of ~4–6 m.

The digital oscilloscope used in 2007/8 and 2010/11 had a sampling rate of 100 MHz, giving a time resolution of 10 ns, equivalent to 1.68-m vertical resolution on ice. The sample rate in 2013/14 was 250 MHz, giving a 4-ns time resolution equivalent to 0.67-m vertical resolution. The system was therefore capable of detecting a change in ice thickness from trace to trace of <2 m when a high-amplitude bed reflection was present.

**Table 2 Radar-track crossover statistics**

| | Number of crossovers analysed | Mean ice-thickness difference/m | Standard deviation of ice-thickness difference/m | % age of crossovers where along-flow elevation <across-flow elevation |
|---|---|---|---|---|
| iSTARt6 | 11 | 2.1 | 1.1 | 100 |
| iSTARt1 | 36 | 9.0 | 7.1 | 84 |
| 2007t1 | 9 | 6.1 | 2.7 | 89 |
| iSTARt5 | 22 | 2.7 | 2.6 | 100 |
| 2010tr | 1 | 7.2 | n/a | n/a |
| 2007tr | 30 | 8.3 | 3.7 | 100 |
| iSTARt7 | 27 | 3.2 | 2.8 | 52 |
| iSTARit | 18 | 8.7 | 8.2 | 44 |
| iSTARt9 | 15 | 15.2 | 7.2 | 80 |

Crossover locations as depicted in Fig. 4. The largest discrepancies (6–15 m) occur at the patches with greater topographic variability at the crossover locations (iSTARt1, iSTARit and iSTARt9) or acquired in the earlier seasons (2007t1, 2007tr and 2010tr), when the radar oscilloscope had a lower sampling rate as described in the Methods summary. In all cases except the non-streamlined intertributary patch iSTARit, the majority of ice-thickness measurements acquired when driving orthogonal to flow exceeded those acquired when driving along flow. This is an expected result over streamlined topography as the radar footprint will capture the flanks of adjacent ridges (e.g., MSGL crests) when driven along tracks paralleling ridges, and will be more pronounced where the amplitude of streamlined topography is greater

For all radar surveys in 2010/11 and 2013/14, the radars were driven along pre-planned lines oriented orthogonal to the ice flow, spaced 500-m apart, following the practice previously applied to surveys on Rutford Ice Stream[34, 45] and Carlson Inlet[46]. In each of the survey 'patches', extra radar profiles acquired at different orientations (usually acquired opportunistically in transit between a camp and starting a new cross-flow radar profile) were used as an additional check for consistency in data acquisition across the patch (Fig. 4; Table 2). The radar surveys acquired in 2007/08 had sparser line spacing (1 km for 2007tr, 3 km for 2007t1) but benefitted from multiple along-flow profiles (Fig. 4).

Before radar processing, the dual-frequency GPS data collected during each survey were processed using the Canadian Spatial Reference System (CSRS) Precise Point Positioning (PPP) service (Canadian Geodetic Survey; https://webapp.geod. nrcan.gc.ca/geod/tools-outils/ppp.php). This is an online service that uses the precise GNSS satellite orbit ephemerides to produce corrected $x$, $y$, $z$ coordinates of a constant 'absolute' accuracy. We submitted all data in RINEX format, and processed them in kinematic mode in the International Terrestrial Reference Frame. Each output was provided in WGS84 coordinates, with $z$ values given with respect to the WGS84 ellipsoid.

For a few radar profiles acquired in 2013/14, comprising small proportions of the data collected across three of the survey patches (iSTARt1, iSTARt5 and iSTARt6), dual-frequency GPS files were not collected or were overwritten due to a GPS software malfunction. For the affected radar profiles (marked on Fig. 4), we used $x$, $y$ coordinates acquired by the single-frequency GPS during the same drive. Cross-checking between dual- and single-frequency GPS-acquired coordinates for all other profiles where both the dual- and single-frequency systems were working showed that the single-frequency GPS-acquired $x$, $y$ coordinates were broadly comparable with the dual-frequency GPS data and had an accuracy of <1 m (confirming their usability) but that the single-frequency GPS $z$ coordinates had a vertical accuracy of $\pm 10$ m (confirming their non-usability). For the affected radar profiles, we therefore recovered $z$ from surface elevation data generated from DigitalGlobe WorldView-1 and WorldView-2 along-track stereo image data[47]. For each of iSTARt1, iSTARt5 and iSTARt6, we first generated two DEMs: one a weighted average of WorldView-generated DEMs from 2010 to 2015 (hence spanning our 2013/14 survey), and the other a 50-m-resolution gridded output of the CSRS-PPP-processed surface elevation data generated from the radar profiles where the dual-frequency GPS had worked (which was the majority of radar profiles for each survey patch). From these two products for each survey patch, which showed a maximum elevation difference of 3 m but usually ~1 m, we generated difference maps from which we recovered $z$ coordinates to fit to the single-frequency $x$, $y$ coordinates for the affected radar profiles.

**Radar data processing.** All the radar data were processed using ReflexW software (Sandmeier Geophysical Research). We first assigned positioning information to every recorded radar trace from the CSRS-PPP output (or its substitute data from WorldView DEMs) as described above. We suppressed noise induced by the arrival of the direct wave using an average filter, and then used a band-pass filter to increase the signal-to-noise ratio. The data were then given an amplitude-scaling proportional to the two-way travel time to compensate for spherical spreading of the radar wavefront with depth, and were migrated using the Kirchoff function to reassign energy back to source points. Finally, a further band-pass filter removed processing-induced noise. The data were archived in SEG-Y format.

To pick the ice-bed interface and to generate the DEMs for each radar-survey patch, the SEG-Y files were imported into the Schlumberger Petrel interpretation software suite. The travel times of each reflection from the ice-bed interface were picked on the radar profiles using a semi-automatic picker that followed the onset of the bed-reflection wavelet from trace to trace. In every radar trace collected for this study, the bed-reflection wavelet had a significantly higher reflection amplitude than nearby internal layers and with a negligible signal below, and because the radar-acquisition strategy involved the interpretation of multiple parallel radar tracks, consistency in identifying the bed could be checked profile by profile: for both of these reasons, we are confident that the bed returns could be picked with high precision, and were not affected by the quarter-wavelength criteria that control the resolution between two adjacent reflectors of similar amplitude. The procedure created a raw data set of $x$, $y$, $t$ coordinates, where $x$ and $y$ were the coordinates in South Pole stereographic projection, and $t$ was the two-way travel time. We converted $t$ into depth $h$ using a single value of radiowave speed through ice of 0.168 m ns$^{-1}$, and adding 10 m to correct for propagation of the radiowaves through a layer of firn at the surface, based on prior experience[34, 45]. The elevation values for the bed at each radar trace were then calculated by subtracting $h$ from the surface elevation, giving bed elevations with an estimated vertical precision of $\pm 3$ m. We acknowledge that using the single value for radiowave propagation is a standard simplification that neglects possible, but unquantifiable, variability in attenuation at each survey patch, which may arise due to variations in ice temperature or chemistry; however, we argue that making such an assumption is reasonable because the relatively local areas covered by each survey patch. Varying the radiowave speed across a range of 0.167–0.170 m ns$^{-1}$, a plausible range based on reported values across Antarctica[48, 49], has the effect of varying ice thickness (hence bed elevation) across a range of ~35 m for ice between 1.5 and 2-km thick. This does not have an impact on the essential shapes of the bed topography maps per survey patch, and affects the trough-crest heights of MSGL (Table 1) by <10%. Our use of a single firn correction value is also adopted because we cannot quantify variations in firn properties along each radar track: this is a possible source of error in the absolute values of bed elevation data that we report. However, because our data collection strategy involved acquiring multiple parallel profiles, so that consistency of radar returns between adjacent profiles could be ascertained, we argue that any such firn-derived errors most likely affect adjacent tracks consistently, and therefore do not affect the main findings of our study.

**Deriving digital elevation models and bed roughness.** To produce the images of the bed for Figs. 1 and 2 of this paper, we interpolated the measured $x$, $y$, $z$ bed elevations onto a 40 m (cross-flow) × 100 m (along-flow) grid oriented orthogonal to the profile lines using a natural neighbour algorithm with a 5:1 anisotropy ratio aligned along the ice flow direction. This interpolation scheme preserves the continuity of the features that are elongate in the ice-flow direction while preserving some of the high spatial sampling along the cross-flow radar tracks.

Bed roughness for Table 1 and Fig. 3 was calculated using a forward fast Fourier transform (FFT) technique described in ref. [50]. The code enables the derivation of roughness along 2D tracks using moving windows of predefined length. We applied the code along 2-km moving windows, which are sufficiently long to capture multiple streamlined landforms when applied across flow, yet small enough to capture roughness variability within each of the survey patches.

**Data availability.** All the data in this paper are available from the lead author by request and will be made available on the NERC/iSTAR GIS site, http://gis.istar.ac.uk/.

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

## Acknowledgements

This work was supported by funding from the UK Natural Environment Research Council (NERC) iSTAR Programme (Grants NE/J005665 and NE/K011189), NERC grants NE/B502287/1 and NE/J004766/1 and the British Antarctic Survey (BAS) Polar Science for Planet Earth Programme. D.E.S. was supported by a NASA NESSF fellowship (NNX12AN36H). Bathymetric data used for Fig. 2c and f were sourced from the Bolin Centre Database Oden Mapping Data (cruise OSO 0910; http://oden.geo.su.se/oso0910) and NSF/IEDA Marine Geoscience Data System (http://www.marine-geo.org/tools/search/Files.php?data_set_uid=20080) respectively, and we thank the lead authors M. Jakobsson and F.O. Nitsche for their lodging. All fieldwork was supported by the staff at BAS's Rothera Research Station and members of the iSTAR Traverse. In particular, we thank James Wake, Tim Gee, Jonny Yates (2013/14), David Routledge (2010/11) and Feargal Buckley, Chris Griffiths and Julian Scott (2007/08) for their help with data

acquisition. We thank three anonymous referees for thorough and constructive reviews, which improved the final form of the manuscript.

## Author contributions

R.G.B. and D.G.V. designed the research and wrote the paper. Data from 2013/14 were acquired by D.D., D.G.V., S.L.C., R.G.B., A.M.S. and J.D.R. Data from 2007/08 to 2010/11 were acquired by R.G.B. and E.C.K., respectively, under research programmes led by A.M.S. All radar data were processed by R.G.B., D.D. and E.C.K.; O.J.M. contributed to processing of radar data from 2007/08; and D.E.S. provided high-resolution surface ice DEMs to calibrate data acquired in 2013/14. All authors contributed ideas and edits to the manuscript.

## Additional information

**Competing interests:** The authors declare no competing financial interests.

