## [Peer Review File · Nature Communications]

Reviewers' comments:

Reviewer #1 (Remarks to the Author):

This paper provides an excellent description of important new data on bedform under Pine Island Glacier. The level of detail is completely new, for the areas surveyed. There are important implications and connections of this new data for ice sheet models, especially for projections of future retreat of Pine Island and similar glaciers. These implications are discussed very well and appropriately in the paper. Overall the paper is well written, clear and succinct, and the importance of the data to larger issues of ice sheet retreat and sea level rise makes the paper suitable for Nature Communications. I have no significant criticisms, and think it is suitable for publication essentially as is. The first two suggestions below could be added at the discretion of the authors.

1. The numerical values provided for each survey patch in Fig. 1 are very helpful. Would it be possible to also provide some values representing the dominant vertical amplitude, and horizontal scales of the features for each patch. I realize that this cannot be thoroughly captured by 2 or 3 numbers, but some rough values would be helpful I think.

2. I suggest showing all these values (in Fig. 1, and suggested above) in a table, with columns for the various values, and rows for patches d to l. And/or a couple of plots showing pair-wise correlations between beta vs. roughness, and thinning propagation vs. roughness. These would back up qualitative statements about such correlations in the text (lines 105-106, 128-129).

3. As discussed, knowledge of detailed bed forms as here will be very valuable and perhaps necessary for projecting ice retreat in other ice streams of Antarctica. Line 134 mentions "the urgent need for techniques for efficient measurement...". Is it possible to provide further detail and discuss explicit measurement techniques that could provide this data? Another aspect of course is the type of bed, hard bedrock vs. deformable sediments? Is there independent explicit data in this region to complement the discussion (lines 78-91) that suggests deformable beds correspond to MSGL-scale features, and harder bedrock to rougher topography?

4. The role of the two "analogous" patches from refs. 26 and 28 in Fig. 2c,f is not clear to me. Are these intended to illustrate the best available data from previous work, comparable resolution to here, but just for individual patches, not with the wider coverage provided here? This could be clarified briefly in the caption.

5. Could "upstream propagation rate of ice stream thinning" be defined in the Fig. 1 caption, perhaps just by specifying the threshold value used for thinning vs. not thinning?

Reviewer #2 (Remarks to the Author):

This paper presents a novel and highly detailed dataset of subglacial conditions underneath one of the most emblematic glaciers of the Antarctic ice sheet. Until now, such detailed imprints of ice

sheet/bed interaction, such as mega scale glacial lineations (MSGs) were only witnessed in deglaciated areas on the continental shelf and observed from ships. The detailed radio-echo sounding survey carried out by Bingham et al. in challenging terrain reveals for the first time similar detailed features underneath the active ice sheet. A dataset with such stunning detail on the subglacial characteristics is a game changer in glaciology and opens up new perspectives on understanding the control of basal conditions on ice sheet and ice stream behavior. While the authors do a splendid job on presenting the dataset, the subsequent analysis and consequences for ice sheet modeling has yet to be demonstrated.

Half of the paper concerns potential impacts for ice sheet modeling and the authors compare their results with previous model inversions carried out on coarse grid scales using the - in this case obsolete - BEDMAP2 dataset. Obviously this is like comparing apples and oranges. This is quite astonishing, since co-authors Cornford and Arthern are ice sheet modelers and potentially could employ the new dataset to carry out detailed inversions at high resolution (BISICLES for instance can cope with grid sizes of the order of hundreds of meters) and analyse the link between friction coefficients and topographic features as well as the impact of new inversions on future behavior of Pine Island Glacier (although the latter maybe good for a separate paper). This can perfectly be done on a basin scale and does not require pan-Antarctic mapping.

Subglacial imprints and bed conditions are transferred to the surface through ice dynamics (and membrane stresses within the ice mass). Therefore, small wavelength features are quickly damped resulting in a relatively smooth surface and smooth surface velocity. Previous model attempts had to deal with a surface expression of high detail (and now Sentinel data produce surface velocity maps at unprecedented detail and quality, e.g., Mouginit et al.) but with a bed topography of much coarser resolution. The new dataset changes this and the authors should demonstrate this quantitatively instead of making a meager comparison with old data. They could then compare the new inversions with those obtained from Arthern et al (2015). If new model inversions fail to reproduce a detailed basal friction field in agreement with the type of features observed (i.e., low friction across MSGs, high friction across rough bed topography), then something can be said on the lack of understanding of basal processes and their representation in ice sheet models, but I am not sure that this would be case. Nevertheless, the current analysis doesn't proof anything on the quality of ice sheet models, but emphasizes our general lack of knowledge on basal topography, the quality of previous model results without necessarily pointing to our lack of understanding of basal processes.

Detailed remarks

Lines 48-51: I agree that the magnitude and rate of projected contribution depends on the pace at which the retreat may propagate, but is this largely controlled by poorly constrained basal boundary conditions? Model inversions do produce basal traction maps that are in general agreement with the new topographic features (albeit on a coarser scale), but most of the uncertainty in projections is related to the understanding of ice shelf buttressing/calving and their effect on grounding line dynamics (and control of marine ice-sheet instability).

Reviewer #3 (Remarks to the Author):

Review of: Diverse landscapes beneath Pine Island Glacier influence rates of ice loss, Bingham et al., 2017

A. Summary of Key Results

The key results claimed in this paper are (i) that the bedrock topography beneath Pine Island Glacier (PIG) is rougher than previously thought (L28), and (ii) that this roughness is great enough to have a major impact on ice flow (L29). Further to this, the paper argues that the impact of the

rough bed topography on ice flow is so great, that it (iii) influences the rate of ice loss (L2), and therefore (iv) the future ice sheet evolution and projected sea level rise contribution (L30) from this sector of Antarctica.

This paper does present a new high resolution bed topography dataset over patches covering ~15% of the PIG drainage basin, collected using a robust and technically sound method. This supports their first key finding (i) that bed roughness in the study areas is greater than the existing alternative coarser resolution datasets. However, my major criticism of the work is that the authors largely rely on a discussion of existing theory, with little quantitative analysis to support their other key findings; that the high resolution bed topography has an impact on ice flow (ii), or that if incorporated into an ice sheet model that it would influence the rate of ice loss (iii) or future sea level contribution (iv). I don't know if key numbers to support these 3 conclusions have not been provided because they have been omitted, or because the new bed topography dataset is either not rough or spatially extensive enough to cause the changes in flow etc that are alluded to. However, aside from a new dataset description, most of the novelty of this paper hinges on the high resolution bed topography having a direct influence on ice flow, the rate of ice loss, and future sea level rise contribution. Unless the paper is revised to clearly provide quantitative information to support these conclusions, then I am not convinced that the paper provides robust evidence to support these major claims.

B. Originality and interest

The bedrock topography measurements collected on PIG are new, and provide detail about the spatial variability of subglacial topography in this area that has not been previously observed. While the range of topographic variability (up to 700m) is much smaller than new 'Alpine' mountain ranges (1000's m variability) measured elsewhere in Antarctica (Fretwell et al., 2016), I agree with the authors statement that understanding the sub-glacial topography of fast flowing ice streams that are out of balance is arguably more important because of the implications it may have for future stability. Alternative, more spatially extensive estimates of bed topography can be made by inverting from ice flow, however there is clear value in having a direct in situ measurement, as presented in this study. The dataset will be of most interest to ice sheet modelers, but it will also be interesting for other glaciologists who may wish to interpret other observations in relation to the bed conditions.

C. Data & methodology: validity of approach, quality of data, quality of presentation

The method used to collect the in situ bedrock topography data is sound and has been previously demonstrated in other study areas. The ice sheet modelling is also technically sound, with a detailed methodology already fully reported in previous publications. I agree with the authors decision to reference this previous work rather than repeating the methods description. The new bed topography dataset has been very clearly presented in these figures, bar some minor editorial comments. I do think the paper is lacking a figure or table to support the other 3 key points of the paper (ii, iii, iv) that are the main focus of the abstract.

Figure 1.

- It is awkward to compare the model output numbers for each image patch when listed alongside the image. I would prefer this information to be in a separate table, alongside equivalent numbers retrieved using the coarse bedrock topography to constrain the model. This would more clearly demonstrate the improvement offered by the new bedrock topography.
- Coordinates should be included on each image patch.
- The 5km xy scale bar is difficult to use due to the 3D orientation
- I don't think location map 1.b is needed.
- The orientation of the image patches according to ice flow rather than geographic location makes it difficult to locate them on the map (F1c).

Figure 2.

- There is no color bar so it's not possible to interpret the data.
- My comments on the coordinates and scale bar are the same as for Fig1.

- The location of F2c and f is not on the main location map in F1, this should be added. However, as F2c and f aren't referenced in the main manuscript, I don't think they are that relevant to the paper?

- I think it would be more useful for this figure to show a set of 9 difference maps between the new high and old low resolution data as it would enable the improvement in all study areas to be shown.

Figure 3.

- There is no color bar.

- There are no coordinates.

- The majority of the information presented in this figure is straight duplication of information from figure 1. Fig 3a-c can be removed completely, and I would prefer the spatially variable uncertainty map to be shown behind the ground track locations as this is not already presented elsewhere.

D. Appropriate use of statistics and treatment of uncertainties

The paper is lacking quantitative statements to back up many of its key conclusions, however I have made this point elsewhere in the review.

There is no error budget provided for any of the key numbers presented in the main manuscript of this paper. As one of the main results of this paper is improved knowledge of the bed topography, it's important to establish if any of the observed small scale variability could be accounted for by measurement uncertainty. It may be challenging to generate a fully quantitative error budget, but at a minimum a discussion of the sources of error should be included in the methods description.

A spatially variable uncertainty map should be produced for each of the 9 patches, and the method used to determine this uncertainty should be outlined. As the background data presented in methods Figure 3 is duplication of the data already presented in Figure 1, I suggest the uncertainty maps could be presented here instead.

As stated in section 'C', I would like to see model outputs for each site derived from both the coarse and high bedrock topography, as this would clearly demonstrate the improvement offered by the new bedrock topography.

E. Conclusions: robustness, validity, reliability

As I stated in section 'A', in its present form I don't think this paper presents enough quantitative evidence to justify its title, or to support 3 of the major conclusions of the paper stated in the abstract. It is not clear to me if this is because model runs using the new high resolution data don't provide this evidence, or because it was just omitted from the text. Without this evidence the importance of the paper to both specialists in the field and the wider community is greatly reduced as the paper is simply describing a new dataset. If the authors have already done the analysis to support these conclusions, but simply omitted to include the numbers in this version of the paper, then it will be very easy for them to improve the manuscript. Conversely, if they haven't, it may be impossible within a reasonable timeframe.

F. Suggested improvements: experiments, data for possible revision

The following list states detailed, specific edits that should be made to the manuscript:

L17 – 30 – The abstract does not contain any quantitative details about the key results of this study. For example, what is the range of bed topography measured in this new dataset; what is the percentage difference between this an alternative coarse resolution dataset; what is the change in friction or form drag simulated by an ice flow model after using the new high resolution bed topography; what is the change in ice flow or ice loss simulated after incorporating this new information? The manuscript would be significantly improved if the authors were to provide specific key numbers in the abstract as it would provide facts to support all their key arguments. At the moment, all the statements in the abstract are qualitative which makes it impossible for the reader to get an immediate impression about the significance of this work.

L38 – Basal boundary conditions are a source of uncertainty, but the authors could provide broader context in the introduction by listing other sources of uncertainty. A citation should be provided to support the statement that basal boundary conditions are one of the greatest.

L42 – References for the 'few' comparable observational data should be provided, along with a specific mention in the manuscript if they were also collected on PIG as this may have implications for the novelty of the dataset reported in this paper.

L43 – State the horizontal and vertical spatial resolution of the bed topography data, 'high' is generic.

L58 – Were only 9 patches collected, or was it more? If so what is the reason for excluding the additional data?

L61-62 – Provide statistics on the mean and variability of the difference between the new data and the old coarse data. Even a range would be helpful as 'diverse' could mean different things to different readers.

L67 – How do these observations confirm that ice flow hasn't changed for thousands of years? Could it not just as easily be an order of magnitude longer? Unless the authors are confident about the facts in this statement, I'd remove it as I think it's speculative, and it isn't needed to support the main conclusions they are trying to draw.

L70 – State what distances are classed as low and high wavelength features.

L88 – Rephrase the wording of 'aerogravity high'. Aerogravity is the technique so probably doesn't warrant a mention; and the word 'high' is misleading within a discussion of bedrock topography, when I think the authors are describing roughness/variability, rather than a high elevation feature.

L102 – What is the spatial resolution of the ice sheet model (or range if variable grid used), and what is the minimum spatial resolution of bed topography features that it can resolve? How does that compare with the spatial resolution of the bed topography features in this new dataset?

L105-107 – I don't think this statement is clear. What is the traction coefficient derived at each site from coarse resolution bed topography (BEDMAP2), vs the same coefficient derived from the new high resolution bed topography.

L109 – Is unresolved bed topography the only explanation for variability in basal traction? I would prefer to see a little consideration of other possible factors.

L115 – 125 – I agree with the authors that this new dataset has advantages over BEDMAP2, however I don't think 'significant and influential' has been demonstrated. The key point the authors make in the title of the paper, is that the high resolution bedrock topography 'influences the rate of ice loss'. For this statement to be proven, I would expect to see some numbers in this paragraph demonstrating how the ice sheet model run with old coarse and the new high resolution bed topography changes the projected rate of ice loss on this ice stream, and the expected contribution to sea level rise.

L192-194 – What are the statistics for the variability of the bed elevation measured at the track crossover points?

L325 – What was the window size used for the each filtering step, and what are the thresholds used to exclude data from the average?

L342 – A single radiowave speed is unlikely to be correct, as acknowledged by the authors (L346). As the spatial variability of relevant parameters throughout the whole ice column isn't known, I suggest a simple sensitivity test to vary the travel speed within a realistic range would be a reasonable way to constrain the impact this may have on the bed elevation measurements.

L342 – Firn depth is known to be spatially variable across PIG, and estimates of this can be easily retrieved from regional climate models, therefore a spatially variable firn correction should be applied. While the author's later state that firn variability along each track isn't known (L350), this argument doesn't hold for firn variability between the data patches. I do appreciate that this point has been addressed and discussed in the methods, so although I know a variable correction could be applied, I'd be happy to accept their arguments if the other reviewers don't consider it a major issue.

G. References: appropriate credit to previous work?

The reference list is good with no major omissions to the best of my knowledge. My only minor

comments are a) for the authors to consider the few specific reference requests made in section F above; and b) that in the introduction, broad sweeping statements are made alongside up to 6 citations. I suggest the authors consider whether all these references are really required to support the individual statement, or would a more focused selection be clearer?

H. Clarity and context: lucidity of abstract/summary, appropriateness of abstract, introduction and conclusions.

The text is well written in coherent sentences with no obvious grammatical errors. This makes the paper clear and easy to read, and the authors should be commended for this. My major criticism of the manuscript is that it lacks quantitative specifics throughout, with the exception of the bed topography section (L53-L91). In particular, the abstract must be revised to contain a number generated during this study that supports each of their 4 key results. Without this, I don't think the authors can claim that they have presented strong evidence to support their conclusions. Specific examples of a qualitative adjective that could easily be replaced with a quantitative fact elsewhere in the paper, are anywhere where the words 'significant', 'major', 'enormous' and 'dramatic' are used.

We are thankful that all three reviewers emphasised and highlighted the importance of the data and the unprecedented high detail they provide across one of the world's least accessible environments. Indeed, while Reviewer#1 noted that the “level of detail is completely new, for the areas surveyed,” we would go further in emphasising that this level of detail, both in terms of its geographical spread (15% of a current ice stream) and its sub-km resolution beneath currently glaciated terrain, is *unprecedented beneath glaciated terrain anywhere*. We have now added a statement in Lines 59-62 to underscore this point more explicitly. Moreover, to clarify a query posed by Reviewer #3, the data presented here represent **all** such data *ever* acquired over Pine Island Glacier since ground radar surveys commenced there in 2007 – we have taken the ground radars there in a total of three austral field seasons and *in this paper we are presenting our entire dataset from Pine Island Glacier, with none of these data having been published anywhere before*. By potentially publishing these data here (a “game changer in glaciology” – Reviewer#2), alongside a clear demonstration that short-wavelength topography is highly likely to matter for ice-retreat projections, we consider that this paper is likely to be very highly cited.

(As an aside, although in this paper we have focussed upon the importance of the observations for future ice-sheet modelling and projections, we are also certain that this first publication of unprecedented detail on a *contemporary glaciated landscape* will also be highly cited by the international landscape-evolution/geomorphology communities.)

We have taken on board the point highlighted by all three reviewers' comments that the initial submission did not take all of its opportunities to present elements of quantitative support to boost its line of reasoning. We have addressed this by adding a new Table 1, a new Figure 3, and referring to the details therein throughout the revised text (e.g. Lines 72-82, 111-114, 132-136). As a key component we have added in calculations of basal roughness across each of the survey patches (Table 1, Lines 194-198, 411-415). We have also added in further details of possible errors (radar-track crossover statistics in a new Table 2 and Lines 394-398) as per the request of Reviewer#3. Further details are given in the detailed responses to reviewers below.

Reviewers #2 and #3 both additionally offered the view that the paper might be improved by incorporating some further elements of ice-sheet modelling. What each of them proposed is subtly different, and we respond in detail on their comments below. In principle, though, we strongly disagree that further modelling beyond what we have already undertaken is necessary for *this* paper. To reiterate, we consider that the purpose of this study is to present to the wider community the new observations, and in so doing to demonstrate (making use of the recent state-of-the-art model results of Arthern et al., 2015; our ref. 37) that such hitherto unimaged high-resolution basal variability is almost certainly affecting rates of ice-stream retreat. In effect, with this paper we are thereby posing a challenge to the ice-sheet modelling community to prove that the various models that have hitherto relied on unrealistically smooth beds still provide adequate bases for projection. To address this challenge robustly can only be done through an intercomparison exercise with many models involved and that is beyond the scope of this study. It is a clear next stage – indeed, and some of our team are actively pursuing it (see response to Reviewer#2 below) – but it is not possible to engage the wider community in this exercise without this paper being published as a first step.

The above has outlined our main actions and position; we now address each of the reviewers' comments point-by-point:

Reviewer #1 (Remarks to the Author):

This paper provides an excellent description of important new data on bedforms under Pine Island Glacier. The level of detail is completely new, for the areas surveyed. There are important implications and connections of this new data for ice sheet models, especially for projections of future retreat of Pine Island and similar glaciers. These implications are discussed very well and appropriately in the paper. Overall the paper is well written, clear and succinct, and the importance of the data to larger issues of ice sheet retreat and sea level rise makes the paper suitable for Nature Communications. I have no significant criticisms, and think it is suitable for publication essentially as is. The first two suggestions below could be added at the discretion of the authors.

We thank Reviewer#1 for his/her extremely positive overview of our paper, and his/her very helpful suggestions.

- 1. The numerical values provided for each survey patch in Fig. 1 are very helpful. Would it be possible to also provide some values representing the dominant vertical amplitude, and horizontal scales of the features for each patch. I realize that this cannot be thoroughly captured by 2 or 3 numbers, but some rough values would be helpful I think.*
- 2. I suggest showing all these values (in Fig. 1, and suggested above) in a table, with columns for the various values, and rows for patches d to l. And/or a couple of plots showing pair-wise correlations between beta vs. roughness, and thinning propagation vs. roughness. These would back up qualitative statements about such correlations in the text (lines 105-106, 128-129).*

We have taken up all of these very useful suggestions. We have retained the annotated values on **Figure 1** but introduced a new table, **Table 1**, which lists all the parameters discussed in the initial submission but adds in further parameters per survey patch as suggested by the reviewer, namely representative horizontal and vertical dimensions of the main features, and also roughness metrics for each patch (the table caption, **Lines 185-198**, gives the details). We have also created a new **Figure 3** showing the pair-wise plots suggested by the reviewer which do, indeed, provide stronger support for the correlations mentioned in the text (**Lines 111-114, 132-136**).

- 3. As discussed, knowledge of detailed bed forms as here will be very valuable and perhaps necessary for projecting ice retreat in other ice streams of Antarctica. Line 134 mentions " the urgent need for*

techniques for efficient measurement...". Is it possible to provide further detail and discuss explicit measurement techniques that could provide this data? Another aspect of course is the type of bed, hard bedrock vs. deformable sediments? Is there independent explicit data in this region to complement the discussion (lines 78-91) that suggests deformable beds correspond to MSGL-scale features, and harder bedrock to rougher topography?

The best prospect ahead for recovering similar data more efficiently appears to lie with the Kansas/CRISIS swath radar that is in an early stage of development. We have added a sentence referencing this prospective way forwards to Lines 142-144.

Regarding whether we have independent data on the type of bed (hard/soft) to complement our discussion of what controls the topographical roughness, we state explicitly in the slightly reworded Lines 84-85 that we have also obtained seismic profiles in all of the radar-surveyed patches, and that at every location they reveal deforming sediments (all published in Brisbourne et al., 2017; our ref. 33). This is restated as part of the discussion in Lines 114-116. The seismic data show very few clear interfaces deeper into the bed, so there is little evidence for the deep subglacial structure. Hence, we consider it more likely that where there are subglacial lineations of amplitudes several 10s of m high, they could express the underlying geology with a thin drape of deforming sediments atop. We have added a clarification on this point to Line 93-94.

4. The role of the two "analogous" patches from refs. 26 and 28 in Fig. 2c,f is not clear to me. Are these intended to illustrate the best available data from previous work, comparable resolution to here, but just for individual patches, not with the wider coverage provided here? This could be clarified briefly in the caption.

The purpose of Figs. 2c and f is merely to demonstrate that the deglaciated topography imaged offshore is very similar, when viewed at the same scales, to the new subglacial topographic maps now recovered from below the contemporary ice-stream bed, as per Lines 66-68. We have amended the Figure 2 caption to clarify that both Figs. 2c and f represent 10 x 15 km subsamples of much wider-coverage offshore datasets (Lines 173-174 and 176-177) and, as per advice from Reviewer #3, additionally marked the locations of these patches into Figure 1b.

5. Could "upstream propagation rate of ice stream thinning" be defined in the Fig. 1 caption, perhaps just by specifying the threshold value used for thinning vs. not thinning?

We have clarified in the figure caption (Lines 167-168) that we used Konrad et al.'s (2017) 1.0 m/yr threshold.

Reviewer #2 (Remarks to the Author):

This paper presents a novel and highly detailed dataset of subglacial conditions underneath one of the most emblematic glaciers of the Antarctic ice sheet. Until now, such detailed imprints of ice sheet/bed interaction, such as mega scale glacial lineations (MSGLs) were only witnessed in deglaciated areas on the continental shelf and observed from ships. The detailed radio-echo sounding survey carried out by Bingham et al. in challenging terrain reveals for the first time similar detailed features underneath the active ice sheet. A dataset with such stunning detail on the subglacial characteristics is a game changer in glaciology and opens up new perspectives on understanding the control of basal conditions on ice sheet and ice stream behavior. While the authors do a splendid job on presenting the dataset, the subsequent analysis and consequences for ice sheet modeling has yet to be demonstrated.

Half of the paper concerns potential impacts for ice sheet modeling and the authors compare their results with previous model inversions carried out on coarse grid scales using the - in this case obsolete - BEDMAP2 dataset. Obviously this is like comparing apples and oranges. This is quite astonishing, since co-authors

Cornford and Arthern are ice sheet modelers and potentially could employ the new dataset to carry out detailed inversions at high resolution (BISICLES for instance can cope with grid sizes of the order of hundreds of meters) and analyse the link between friction coefficients and topographic features as well as the impact of new inversions on future behavior of Pine Island Glacier (although the latter may be good for a separate paper). This can perfectly be done on a basin scale and does not require pan-Antarctic mapping.

We thank Reviewer#2 for his/her glowing comments on the data that our paper provides.

Our main position on the request for further modelling to be incorporated into this paper is summarised in the fourth paragraph responding to the Editor's summary above. Here, more specifically, we disagree with the suggestion that we could apply the ice-sheet model BISICLES to the new data in the manner suggested by Reviewer#2. At the spatial resolution now captured by the new bed data, it is only appropriate to apply a full-Stokes model to investigate fully the link between friction coefficients and topographic features. Some of our colleagues on the NERC-iSTAR project are already pursuing this exercise using the data presented here, but those results are being targeted towards a longer-format journal given the space required to discuss the required modelling developments in full. Ultimately, as we have stated, this exercise would have to be repeated with additional suitable models in a fuller intercomparison exercise for the modelling community to be confident that their findings are robust in the face of our new data.

Subglacial imprints and bed conditions are transferred to the surface through ice dynamics (and membrane stresses within the ice mass). Therefore, small wavelength features are quickly damped resulting in a relatively smooth surface and smooth surface velocity. Previous model attempts had to deal with a surface expression of high detail (and now Sentinel data produce surface velocity maps at unprecedented detail and quality, e.g., Mouginit et al.) but with a bed topography of much coarser resolution. The new dataset changes this and the authors should demonstrate this quantitatively instead of making a meager comparison with old data. They could then compare the new inversions with those obtained from Arthern et al (2015). If new model inversions fail to reproduce a detailed basal friction field in agreement with the type of features observed (i.e., low friction across MSGs, high friction across rough bed topography), then something can be said on the lack of understanding of basal processes and their representation in ice sheet models, but I am not sure that this would be case. Nevertheless, the current analysis doesn't prove anything on the quality of ice sheet models, but emphasizes our general lack of knowledge on basal topography, the quality of previous model results without necessarily pointing to our lack of understanding of basal processes.

We completely concur with the reviewer's statement that "the current analysis doesn't prove anything on the quality of ice-sheet models." Indeed, nowhere in the paper have we spoken to the *quality* of ice-sheet models. Our key line of reasoning is to show that the basal traction parameter inverted by most state-of-the-art models corresponds closely to the short-wavelength topographic variability (i.e. bed roughness) shown by the new data, and hence we can now state with confidence that the basal traction (hence ice flow) of Pine Island Glacier is much more heavily influenced by form drag, i.e. as opposed to basal friction, than has previously been shown to be the case.

Detailed remarks:

Lines 48-51: I agree that the magnitude and rate of projected contribution depends on the pace at which the retreat may propagate, but is this largely controlled by poorly constrained basal boundary conditions? Model inversions do produce basal traction maps that are in general agreement with the new topographic features (albeit on a coarser scale), but most of the uncertainty in projections is related to the

understanding of ice shelf buttressing/calving and their effect on grounding line dynamics (and control of marine ice-sheet instability).

It has been shown in time-dependent models that while the initial impetus for change may be ice-shelf thinning and loss of downstream buttressing, the scale of the long-term change is heavily dependent upon the choice of the power law in the basal slip condition. Perhaps the highest-profile recent study to have demonstrated this is Ritz et al. (2015; Nature; our ref. 12, which we have now added into the reference string supporting the relevant sentence (Lines 49-52)).

Reviewer #3 (Remarks to the Author):

A. Summary of Key Results

The key results claimed in this paper are (i) that the bedrock topography beneath Pine Island Glacier (PIG) is rougher than previously thought (L28), and (ii) that this roughness is great enough to have a major impact on ice flow (L29). Further to this, the paper argues that the impact of the rough bed topography on ice flow is so great, that it (iii) influences the rate of ice loss (L2), and therefore (iv) the future ice sheet evolution and projected sea level rise contribution (L30) from this sector of Antarctica.

This paper does present a new high resolution bed topography dataset over patches covering ~15% of the PIG drainage basin, collected using a robust and technically sound method. This supports their first key finding (i) that bed roughness in the study areas is greater than the existing alternative coarser resolution datasets. However, my major criticism of the work is that the authors largely rely on a discussion of existing theory, with little quantitative analysis to support their other key findings; that the high resolution bed topography has an impact on ice flow (ii), or that if incorporated into an ice sheet model that it would influence the rate of ice loss (iii) or future sea level contribution (iv). I don't know if key numbers to support these 3 conclusions have not been provided because they have been omitted, or because the new bed topography dataset is either not rough or spatially extensive enough to cause the changes in flow etc that are alluded to. However, aside from a new dataset description, most of the novelty of this paper hinges on the high resolution bed topography having a direct influence on ice flow, the rate of ice loss, and future sea level rise contribution. Unless the paper is revised to clearly provide quantitative information to support these conclusions, then I am not convinced that the paper provides robust evidence to support these major claims.

We thank Reviewer#3 for his/her thorough and critically constructive review.

We consider that we have now addressed Reviewer#3's criticism of what s/he terms claim (ii) by adding in **Table 1** (listing of metrics) and **Figure 3** (relation of roughness to basal traction/thinning propagation), and using these numbers more within the main text (e.g. **Lines 72-82, 111-114, 132-136**).

Regarding the reviewer's criticism that we do not have the results to support what s/he terms claims (iii) and (iv), we contend that this is based on a misreading of our paper summary **Lines 28-30**. Reviewer #3 writes that "the paper argues that the impact of the rough bed topography on ice flow is so great, that it (iii) influences the rate of ice loss (L2) [sic? – presumably L29], and therefore (iv) the future ice sheet evolution and projected sea level rise contribution (L30) from this sector of Antarctica." However, the sentence (now slightly revised, but, with respect to this comment, effectively unchanged) reads: "We show that these diverse subglacial landscapes impact on ice flow, and present a challenge for modelled ice-sheet evolution, and, in turn, projected global sea-level rise from ice-sheet loss." The key difference that Reviewer#3 has neglected is the underlined phrase, and the reasons why we consider this to be a challenge for ice-sheet modelling (hence sea-level rise predictions) are expanded upon throughout **Lines 99-136** with reference to how published state-of-the-art models work.

B. Originality and interest

The bedrock topography measurements collected on PIG are new, and provide detail about the spatial variability of subglacial topography in this area that has not been previously observed. While the range of topographic variability (up to 700 m) is much smaller than new 'Alpine' mountain ranges (1000's m variability) measured elsewhere in Antarctica (Fretwell et al., 2016), I agree with the authors statement that understanding the sub-glacial topography of fast flowing ice streams that are out of balance is arguably more important because of the implications it may have for future stability. Alternative, more spatially extensive estimates of bed topography can be made by inverting from ice flow, however there is clear value in having a direct in situ measurement, as presented in this study. The dataset will be of most interest to ice sheet modelers, but it will also be interesting for other glaciologists who may wish to interpret other observations in relation to the bed conditions.

We take these comments to reinforce the position stated by all reviewers that the dataset the paper provides is unique and of wide interest, value and application to multiple readers.

C. Data & methodology: validity of approach, quality of data, quality of presentation

The method used to collect the in situ bedrock topography data is sound and has been previously demonstrated in other study areas. The ice sheet modelling is also technically sound, with a detailed methodology already fully reported in previous publications. I agree with the authors decision to reference this previous work rather than repeating the methods description. The new bed topography dataset has been very clearly presented in these figures, bar some minor editorial comments. I do think the paper is lacking a figure or table to support the other 3 key points of the paper (ii, iii, iv) that are the main focus of the abstract.

We hope that the new **Table 1** and **Figure 3** answer to the final sentence here.

Figure 1.

- *It is awkward to compare the model output numbers for each image patch when listed alongside the image. I would prefer this information to be in a separate table, alongside equivalent numbers retrieved using the coarse bedrock topography to constrain the model. This would more clearly demonstrate the improvement offered by the new bedrock topography.*
- *Coordinates should be included on each image patch.*
- *The 5 km xy scale bar is difficult to use due to the 3D orientation*
- *I don't think location map 1.b is needed.*
- *The orientation of the image patches according to ice flow rather than geographic location makes it difficult to locate them on the map (F1c).*

We appreciate the care taken by the reviewer to suggest these improvements to all of the diagrams.

- We have added the table as recommended (**Table 1**) though have retained the parameters τ_b , U_b , P_r and β as annotations to **Figure 1** as well. **Table 1** includes subglacial roughness now recovered along- and across-flow both from the new dataset and the previous coarse bedrock topography, ie. Bedmap2.
- We have annotated the latitude/longitude of the centre of each patch below the patch name in each of **Fig. 1 d-l**.
- We prefer to retain the 5 km scale bar in **Figure 1** as is.
- We retain **Figure 1b**, as we think it provides useful context, and it is now where we have added the locations of Figures 2c and f.

Additionally, we note that all of the radar data and DEMs will be made freely available online upon publication. This will allow readers, for example, to pinpoint all specific coordinates and produce their own perspective views.

Figure 2.

- *There is no color bar so it's not possible to interpret the data.*
 - *My comments on the coordinates and scale bar are the same as for Fig 1.*
 - *The location of F2c and f is not on the main location map in F1, this should be added. However, as F2c - and f aren't referenced in the main manuscript, I don't think they are that relevant to the paper?*
 - *I think it would be more useful for this figure to show a set of 9 difference maps between the new high and old low resolution data as it would enable the improvement in all study areas to be shown.*
- Our apologies for omitting the colour bars – now added (Figure 2, all panels).
 - Patch coordinates added for each panel as for Figure 1; scale bar unchanged (Figure 2, all panels).
 - Locations of Figures 2c and f are now marked on Figure 1b. The reference to panels c and f is in Lines 66-68.
 - We consider that the current Figure 2 panels a versus b and d versus e adequately make the point that the new data change our view of the bed shape.

Figure 3.

- *There is no color bar.*
- *There are no coordinates.*
- *The majority of the information presented in this figure is straight duplication of information from figure 1. Fig 3a-c can be removed completely, and I would prefer the spatially variable uncertainty map to be shown behind the ground track locations as this is not already presented elsewhere.*

It is a fair point that there is repeated information between what is now Figure 4 and Figure 1. The lowermost suggestion made above is to substitute the topographic colourmaps with “spatially variable uncertainty maps” which Reviewer#3 requests below in his/her Section D comments. We argue below that this product is not viable. Other possibilities are to plot ice surface elevation or MODIS/Landsat imagery in place of the bed-topography colourmaps, but in practice they add little to the paper. The overriding motivation for Figure 4 is to make explicit the radar tracks per patch and the locations of crossovers used to address errors (see response to Reviewer#3, L292-294, below). We consider this more in the category of auxiliary information, hence our suggestion that it is an online-only figure. As an online-only figure we seek guidance from the editor as to whether it is worthwhile retaining panels a-c and/or to duplicate the coordinate and colourbar information from Figure 1.

D. Appropriate use of statistics and treatment of uncertainties

The paper is lacking quantitative statements to back up many of its key conclusions, however I have made this point elsewhere in the review.

There is no error budget provided for any of the key numbers presented in the main manuscript of this paper. As one of the main results of this paper is improved knowledge of the bed topography, it's important to establish if any of the observed small scale variability could be accounted for by measurement uncertainty. It may be challenging to generate a fully quantitative error budget, but at a minimum a discussion of the sources of error should be included in the methods description.

A spatially variable uncertainty map should be produced for each of the 9 patches, and the method used to determine this uncertainty should be outlined. As the background data presented in methods Figure 3

is duplication of the data already presented in Figure 1, I suggest the uncertainty maps could be presented here instead.

As stated in section 'C', I would like to see model outputs for each site derived from both the coarse and high bedrock topography, as this would clearly demonstrate the improvement offered by the new bedrock topography.

Reviewer#3 states that we supply no error budget and implies that there will be considerable spatial uncertainty across the patches that we could demonstrate in map form. However, we consider that our methods summary already includes mention of the main errors, and that these are relatively constant across the patches. Firstly, we have explained that the radar system is capable of detecting changes in ice thickness of < 2 m along-tracks (Lines 329-333). Secondly, we have detailed the accuracy of the GPS used for navigation (Lines 342-366). Thirdly, we have given an estimated vertical precision of ± 3 m for picking the bed echoes (Line 390). Reviewer#3, below (comment re L292-294), rightly requests statistics for crossovers, which we have also now added as a new Table 2 (referenced in main text at Line 339).

None of these principal sources of error varies spatially, so cannot be used to draw spatially variable uncertainty maps. The main sources of spatially variable uncertainty would arise from changes to the radar-wave speed through ice and firn, which Reviewer #3 raises w.r.t. L.342 below.

E. Conclusions: robustness, validity, reliability

As I stated in section 'A', in its present form I don't think this paper presents enough quantitative evidence to justify its title, or to support 3 of the major conclusions of the paper stated in the abstract. It is not clear to me if this is because model runs using the new high resolution data don't provide this evidence, or because it was just omitted from the text. Without this evidence the importance of the paper to both specialists in the field and the wider community is greatly reduced as the paper is simply describing a new dataset. If the authors have already done the analysis to support these conclusions, but simply omitted to include the numbers in this version of the paper, then it will be very easy for them to improve the manuscript. Conversely, if they haven't, it may be impossible within a reasonable timeframe.

See our response to Section A.

F. Suggested improvements: experiments, data for possible revision

The following list states detailed, specific edits that should be made to the manuscript:

L17 – 30 – The abstract does not contain any quantitative details about the key results of this study. For example, what is the range of bed topography measured in this new dataset; what is the percentage difference between this an alternative coarse resolution dataset; what is the change in friction or form drag simulated by an ice flow model after using the new high resolution bed topography; what is the change in ice flow or ice loss simulated after incorporating this new information? The manuscript would be significantly improved if the authors were to provide specific key numbers in the abstract as it would provide facts to support all their key arguments. At the moment, all the statements in the abstract are qualitative which makes it impossible for the reader to get an immediate impression about the significance of this work.

We have taken on board the point that further quantification was warranted for the paper in general, and we have introduced specific key numbers into the manuscript (e.g. Lines 71-82). However, we consider the abstract fit for purpose in its current form (albeit with some minor editing from the first submission). To

quote the journal's advice on abstracts, they should "serve both as a general introduction to the topic and as a brief, non-technical summary of the main results and their implications."

L38 – Basal boundary conditions are a source of uncertainty, but the authors could provide broader context in the introduction by listing other sources of uncertainty. A citation should be provided to support the statement that basal boundary conditions are one of the greatest.

Both Joughin et al. (2009) and Ritz et al. (2015) emphasise the point that basal boundary conditions represent one of the greatest sources of uncertainty and are now specifically referenced in support of Lines 37-39.

L42 – References for the 'few' comparable observational data should be provided, along with a specific mention in the manuscript if they were also collected on PIG as this may have implications for the novelty of the dataset reported in this paper.

To our knowledge the only locations surveyed in a similar manner in Antarctica have been on Rutford Ice Stream and Carlson Inlet, as now mentioned in the Methods Summary, Line 336. To reiterate, this paper presents *all* of the radar patches ever collected on Pine Island Glacier, as now made more explicit in Lines 59-62.

L43 – State the horizontal and vertical spatial resolution of the bed topography data, 'high' is generic.

We provide all the information needed as regards the resolution in Lines 55-63 and in the Methods Summary.

L58 – Were only 9 patches collected, or was it more? If so what is the reason for excluding the additional data?

See response to Reviewer#3, L.42, just above.

L61-62 – Provide statistics on the mean and variability of the difference between the new data and the old coarse data. Even a range would be helpful as 'diverse' could mean different things to different readers.

For this sentence, now Lines 64-66, the purpose is primarily to direct the reader to Figure 2, so we do not think that adding the suggested numbers is useful here. Later in the manuscript we have made clear that the "old coarse data" have, at best, 5 km spatial resolution (Lines 117-122).

L67 – How do these observations confirm that ice flow hasn't changed for thousands of years? Could it not just as easily be an order of magnitude longer? Unless the authors are confident about the facts in this statement, I'd remove it as I think it's speculative, and it isn't needed to support the main conclusions they are trying to draw.

The sentence, now Lines 69-71, references two studies that support the statement, and we consider it useful to reinforce the message that the ice-flow configuration across the study catchment has remained spatially stable over a timescale relevant to the likely landscape formation.

L70 – State what distances are classed as low and high wavelength features.

We have now quantified this sentence, Lines 71-75.

L88 – Rephrase the wording of 'aerogravity high'. Aerogravity is the technique so probably doesn't warrant a mention; and the word 'high' is misleading within a discussion of bedrock topography, when I think the authors are describing roughness/variability, rather than a high elevation feature.

We have substituted “aerogravity highs” with “gravity anomalies” (Line 94).

L102 – What is the spatial resolution of the ice sheet model (or range if variable grid used), and what is the minimum spatial resolution of bed topography features that it can resolve? How does that compare with the spatial resolution of the bed topography features in this new dataset?

The sentence (and line being questioned, now Line 107) is describing the *general case* that any ice-sheet model cannot resolve bed topography of wavelength shorter than its (regular or variable grid) resolution. Therefore there is not a single number to be provided with this sentence.

L105-107 – I don’t think this statement is clear. What is the traction coefficient derived at each site from coarse resolution bed topography (BEDMAP2), vs the same coefficient derived from the new high resolution bed topography.

The previous sentence (now Lines 108-111) has stated that the basal traction coefficient derived from inversion (regardless of bed-topographic resolution) necessarily lumps together basal friction and form-drag. This sentence (now Lines 111-113) is stating that variations in the roughness of the newly imaged basal topography strongly match variations in the basal traction coefficient derived from inversion, the point therefore being that, of the two component of basal traction, form-drag is likely to be the main cause of the variation in basal traction across Pine Island Glacier. We specified “one” model because we have only tested this with basal traction derived from our model (Arthern et al., 2015; our ref. 37).

L109 – Is unresolved bed topography the only explanation for variability in basal traction? I would prefer to see a little consideration of other possible factors.

The other principal possible factors being alluded to here are surely changes to the composition and wetness of subglacial sediments, which we have already stated are broadly similar across the PIG catchment according to the seismic measurements (Brisbourne et al., 2017) (Lines 113-116).

L115 – 125 – I agree with the authors that this new dataset has advantages over BEDMAP2, however I don’t think ‘significant and influential’ has been demonstrated. The key point the authors make in the title of the paper, is that the high resolution bedrock topography ‘influences the rate of ice loss’. For this statement to be proven, I would expect to see some numbers in this paragraph demonstrating how the ice sheet model run with old coarse and the new high resolution bed topography changes the projected rate of ice loss on this ice stream, and the expected contribution to sea level rise.

Here we return to our position that the community as a whole would be better positioned to undertake the proposed modelling with multiple models, and that this is a natural follow-up to this paper. We believe that the new Figure 3 plots, which show the relationship between bed roughness from the new topography with basal traction and the upstream-thinning-propagation rates, underscore our use of “significant” in Line 122. We have now preceded “influential” with “potentially” in Line 122.

L292-294 – What are the statistics for the variability of the bed elevation measured at the track crossover points?

This is certainly a relevant addition to the error budget which we had omitted from the first submission. We provide the statistics in a new Table 2, intended as an online-only supplement to the Methods Summary. In a nutshell the crossovers range from ~2-15 m, with the largest discrepancies occurring in areas of high-amplitude streamlining. There is a general systematic trend of thinner ice (by a few m) being measured when driving the radar along flow compared when driving the radar across flow, which is an expected result in areas of streamlined topography as explained in the new Table 2 caption (Lines 428-430).

L325 – What was the window size used for the each filtering step, and what are the thresholds used to exclude data from the average?

This information is all documented with the files we will release with publication of the paper. It could be incorporated into some rather heavy text within the Methods summary, but we prefer simply to direct readers to the relevant readme in the data repository.

L342 – A single radiowave speed is unlikely to be correct, as acknowledged by the authors (L346). As the spatial variability of relevant parameters throughout the whole ice column isn't known, I suggest a simple sensitivity test to vary the travel speed within a realistic range would be a reasonable way to constrain the impact this may have on the bed elevation measurements.

We thank Reviewer#3 for requesting this additional useful sensitivity test. We tested the effect of varying the travel speed between 0.167 and 0.17 m ns⁻¹, a plausible range based on Fujita et al. (2000) and Popov et al. (2003), new references 49 and 50 added to the paper. When using the lower travel speed, ice thickness reduces by 10-12 m in areas of total thickness 1.5-2 km; when using the upper travel speed, ice thickness increases by 20-25 m in areas of total thickness 1.5-2 km. This outcome does not affect the bed topography variability and its relevance as reported for this paper, but it is important information for further possible uses of the data, i.e. cross-referencing with airborne datasets. The most important effect that using different travel speeds might have for *this* paper is whether it affects the heights of the MSGL we have now reported for the new Table 1. We find that at every patch variations in radiowave travel-speed affect the reported heights by <10%.

We have added **Lines 394-398** to the paper to report on the above. Additionally, we note that in the data files to be released with publication of this paper we give the two-way travel times (the raw result), as well as the inferred depth using a radiowave travel time of 0.168 m ns⁻¹. Hence, in effect, the information is available for others to undertake more sophisticated tests with radiowave travel speeds, and firn variability as per the next comment.

L342 – Firn depth is known to be spatially variable across PIG, and estimates of this can be easily retrieved from regional climate models, therefore a spatially variable firn correction should be applied. While the author's later state that firn variability along each track isn't known (L350), this argument doesn't hold for firn variability between the data patches. I do appreciate that this point has been addressed and discussed in the methods, so although I know a variable correction could be applied, I'd be happy to accept their arguments if the other reviewers don't consider it a major issue.

We take Reviewer#3's point; in fact we suspect that unknown firn variability is likely to be the greatest source of error in the absolute values of bed-topography elevation that we report. However, the focus of the paper concerns the within-patch topographic variability, and we do not think that implementing firn corrections affects the findings reported here. It is certainly an immensely worthwhile topic to investigate firn corrections across such a dataset for the future, and in fact this would most robustly be done where shallow radar and ice core data could supplement regional climate model output. With the release of the data files that give the raw two-way travel times, and state explicitly that we have assumed a standard radiowave travel time of 0.168 m ns⁻¹ and a standard firn correction of 10 m, the opportunity will be available to do this, but it is beyond the scope of this paper and, to reiterate, does not affect the paper's principal findings.

G. References: appropriate credit to previous work?

The reference list is good with no major omissions to the best of my knowledge. My only minor comments are a) for the authors to consider the few specific reference requests made in section F above; and b) that

in the introduction, broad sweeping statements are made alongside up to 6 citations. I suggest the authors consider whether all these references are really required to support the individual statement, or would a more focused selection be clearer?

We have added the extra citations as per (a) and considered, but in the end not actioned, item (b).

H. Clarity and context: lucidity of abstract/summary, appropriateness of abstract, introduction and conclusions.

The text is well written in coherent sentences with no obvious grammatical errors. This makes the paper clear and easy to read, and the authors should be commended for this. My major criticism of the manuscript is that it lacks quantitative specifics throughout, with the exception of the bed topography section (L53-L91). In particular, the abstract must be revised to contain a number generated during this study that supports each of their 4 key results. Without this, I don't think the authors can claim that they have presented strong evidence to support their conclusions. Specific examples of a qualitative adjective that could easily be replaced with a quantitative fact elsewhere in the paper, are anywhere where the words 'significant', 'major', 'enormous' and 'dramatic' are used.

As per responses given to previous comments we have added further quantitative facts throughout the paper. The comment certainly alerted us to our overuse of "significantly"! We searched/replaced/reworded appearances of 'significant', 'major' and 'enormous', though have retained some instances where we considered them still appropriate within the context of some sentences/points.

Additional revisions

Finally, we record here a handful of minor revisions we have also included in the resubmission after further discussion between the authors:

Line 17: We replaced 60 with 70 years as this refers to change observed since at least the 1940s.

Lines 49 and 66, new Ref. 20: Added newly published relevant reference.

Line 88 and Ref. 35: Added newly published relevant reference.

Line 108: Rephrased/reordered for improved clarity.

Reviewers' comments:

Reviewer #1 (Remarks to the Author):

The authors have responded thoroughly, including the addition of Table 1 and Fig. 3, showing tabulated quantities for each area. Fig. 3 shows the basic correlations as stated in the text (although with an interesting exception: iSTART6). The importance of the paper for ice-sheet modeling is evident, and I fully accept the author's position that the data itself justifies the paper, and modeling is best left to future assessments by the wider modeling community.

Reviewer #2 (Remarks to the Author):

Bingham and others write in their rebuttal: "In principle, though, we strongly disagree that further modelling beyond what we have already undertaken is necessary for this paper. To reiterate, we consider that the purpose of this study is to present to the wider community the new observations, and in so doing to demonstrate (making use of the recent state-of-the-art model results of Arthern et al., 2015; our ref. 37) that such hitherto unimaged high-resolution basal variability is almost certainly affecting rates of ice-stream retreat." and "In effect, with this paper we are thereby posing a challenge to the ice-sheet modelling community to prove that the various models that have hitherto relied on unrealistically smooth beds still provide adequate bases for projection. To address this challenge robustly can only be done through an intercomparison exercise with many models involved and that is beyond the scope of this study. It is a clear next stage – indeed, and some of our team are actively pursuing it (see response to Reviewer#2 below) – but it is not possible to engage the wider community in this exercise without this paper being published as a first step."

I agree with their comment that the purpose of the paper is to present the new and very intriguing data set. I also agree with the fact that extra model experiments fall outside the scope of this paper. I further agree that going beyond what is presented in the paper will require a high amount of work for the modelling community. However, my major concern remains with the weight that is given to some of the consequences for modelling that are not clearly demonstrated by the analysis in the paper.

They further state that besides the data they give "a clear demonstration that short-wavelength topography is highly likely to matter for ice-retreat projections". It may matter, definitely, but it has not been demonstrated quantitatively, and therefore, the 'highly likely' doesn't fit.

The first thing is the title, where the authors clearly claim that the diverse landscapes influence the rate of ice loss. This is not correct: it shows its potential to influence the rate of ice loss: the diverse landscapes definitely change the basal conditions, but from the modelling that is presented (based on older Bedmap2 data) it is not demonstrated that this will indeed change the rate of mass loss (all depends on how basal conditions influence membrane stresses in the ice sheet and how other potential feedbacks with ice shelves and buttressing come into play). A quantitative measure (apart from roughness and a comparison with previously derived basal friction parameters) is unfortunately not given.

Therefore, the prognostic part in the ice sheet modelling section of the paper (and the title) should be downplayed a bit. This will in no case diminish the importance of the paper, as the dataset presented is already quite attractive.

Below, I list some elements that can aid in making the modelling section more accurate:

Line 105: 'fully three-dimensional ice-sheet model might naturally simulate form drag arising from

basal features longer than its horizontal resolution, but cannot be expected to simulate form drag not represented in the subglacial topographic model on which it rests nor, indeed, due to features of shorter-wavelengths shorter features than it can resolve'.

Such small-scale features will definitely be very difficult to resolve in ice-sheet models, even at the highest resolution. Therefore I would add (to make it sound more positive) that the knowledge on the distribution of the observed features may aid at developing adequate parametrizations and expand our theoretical knowledge on the effect of form drag on ice flow, beyond the scale of valley glaciers.

Line 144: 'Until such independent evaluation of form drag and basal friction is integrated into models, the current generation of ice-sheet models will be hampered in establishing robust projections of ice loss and sea-level rise.'

I agree with hampered, but what is meant by robust? To my knowledge, there are quite a lot of factors that hamper projections of ice loss, and the basal boundary conditions is one amongst many others. That is why ice-sheet modelling actually evolves towards ensembles across parameter uncertainties. Maybe what is needed is theoretical knowledge on how short-wave form drag influences ice flow of large outlet glaciers and ice streams (as opposed to valley glaciers), as I said above.

'such hitherto unimaged high-resolution basal variability is almost certainly affecting rates of ice-stream retreat ': This is also a supposition: for me it is not clear that using old BEDMAP data in modelling part of the paper that the proof of concept is given. The basal variability affects the spatial distribution of basal friction and the basal characteristics, which have an effect on the flow of ice. Whether this influences the rate of ice-stream retreat (in terms of a significant contribution) depends on many other factors as well. Until demonstrated through detailed modelling and comparison, there is no evidence given in the paper.

A novel aspect is definitely that 'hence we can now state with confidence that the basal traction (hence ice flow) of Pine Island Glacier is much more heavily influenced by form drag, i.e. as opposed to basal friction, than has previously been shown to be the case.'

Reviewer #3 (Remarks to the Author):

All 3 reviewers were in agreement that this paper presents an interesting new dataset, collected in challenging terrain, using a technically sound approach. The authors have fully addressed the request from reviewers #R1, #R2 & #R3 for more quantitative information on the height, width and roughness of the bed topography (L72-82 and columns 5 to 8 of Table 1), and the request for more attention to errors from #R3 (Table 2, L394-398). There was consensus amongst all 3 reviewers that the dataset is robust, and described well by this paper.

#R2 and #R3 were in agreement that "While the authors do a splendid job on presenting the dataset, the subsequent analysis and consequences for ice sheet modeling has yet to be demonstrated [#R2]." This was a major concern for reviewers #R2 and #R3, because "most of the novelty of this paper hinges on the high resolution bed topography having a direct influence on ice flow, the rate of ice loss, and future sea level rise contribution [#R3]." Both reviewers #R2 and #R3 requested that the authors conduct additional modelling work to "demonstrate this quantitatively instead of making a meager comparison with old data [#R2]."

The authors have not satisfactorily addressed this request, because they have declined to carry out any additional modeling work. The authors justify this decision because they think "basal variability is almost certainly affecting rates of ice-stream retreat", and they would like a future

publication to "prove that the various models that have hitherto relied on unrealistically smooth beds still provide adequate bases for projection." The title of this paper states that "Diverse landscapes beneath Pine Island Glacier influence rates of ice loss", however as the authors have admitted in their response above, they have not demonstrated that this 'is' the case.

While the new Figure 3a enables the reader to more clearly visualise the relationship between basal traction and observed roughness, its not clear to me that there is a statistically significant correlation between the two parameters, and frustratingly statistics about the strength of the correlation that may support this is once again not provided. There is an appearance of more quantitative evidence in the manuscript due to the addition of Table 2, however the numbers provided in columns 1 to 4 are just copy pasted from the original annotations in Figure 1, therefore don't provide new information to address the concerns of reviewers #R2 and #R3.

This paper remains a good description of a new dataset, that will be a valuable contribution to a future modelling study. However, it is the modelling study that will represent an important advance of significance to specialists in the field: by informing us whether the existing coarse resolution bedrock topography datasets have affected the modelled rate of ice flow, to the extent that the projections of future sea level rise assumed today are in fact incorrect.

Re: Manuscript NCOMMS-17-08366A: "Diverse landscapes beneath Pine Island Glacier..."

Below we detail how we have revised the manuscript. *Text in blue italics* is pasted from reviewer comments received on 5 September 2017; text in black gives our responses, within which **red text** denotes line-number locations of the revisions in the *tracked-changes version of the revised paper*.

Reviewer #1 (Remarks to the Author):

The authors have responded thoroughly, including the addition of Table 1 and Fig. 3, showing tabulated quantities for each area. Fig. 3 shows the basic correlations as stated in the text (although with an interesting exception: iSTART6). The importance of the paper for ice-sheet modeling is evident, and I fully accept the author's position that the data itself justifies the paper, and modeling is best left to future assessments by the wider modeling community.

We thank Reviewer #1 for the endorsement, and indeed the advice from the first review. No further action required.

Reviewer #2 (Remarks to the Author):

I agree with their comment [in our 18 August 2017 rebuttal letter] that the purpose of the paper is to present the new and very intriguing data set. I also agree with the fact that extra model experiments fall outside the scope of this paper. I further agree that going beyond what is presented in the paper will require a high amount of work for the modelling community. However, my major concern remains with the weight that is given to some of the consequences for modelling that are not clearly demonstrated by the analysis in the paper.

They further state that besides the data they give "a clear demonstration that short-wavelength topography is highly likely to matter for ice-retreat projections". It may matter, definitely, but it has not been demonstrated quantitatively, and therefore, the 'highly likely' doesn't fit.

The first thing is the title, where the authors clearly claim that the diverse landscapes influence the rate of ice loss. This is not correct: it shows its potential to influence the rate of ice loss: the diverse landscapes definitely change the basal conditions, but from the modelling that is presented (based on older Bedmap2 data) it is not demonstrated that this will indeed change the rate of mass loss (all depends on how basal conditions influence membrane stresses in the ice sheet and how other potential feedbacks with ice shelves and buttressing come into play). A quantitative measure (apart from roughness and a comparison with previously derived basal friction parameters) is unfortunately not given.

Therefore, the prognostic part in the ice sheet modelling section of the paper (and the title) should be downplayed a bit. This will in no case diminish the importance of the paper, as the dataset presented is already quite attractive.

We thank the reviewer for his/her further time in re-reviewing the manuscript. There is nothing in here with which we fundamentally disagree, and indeed these are very helpful statements for improving the paper's overall balance between innovative data and message.

To address the reviewer's fair and well-expressed concern with the title, we have now restated it as: "Diverse landscapes beneath Pine Island Glacier influence ice flow and potentially rates of ice loss" (**Lines 1-2**).

Alternatively, we would be happy to entitle the manuscript simply “Diverse landscapes beneath Pine Island Glacier influence ice flow.”

Below, I list some elements that can aid in making the modelling section more accurate:

Line 105: 'fully three-dimensional ice-sheet model might naturally simulate form drag arising from basal features longer than its horizontal resolution, but cannot be expected to simulate form drag not represented in the subglacial topographic model on which it rests nor, indeed, due to features of shorter-wavelengths shorter features than it can resolve'.

Such small-scale features will definitely be very difficult to resolve in ice-sheet models, even at the highest resolution. Therefore I would add (to make it sound more positive) that the knowledge on the distribution of the observed features may aid at developing adequate parametrizations and expand our theoretical knowledge on the effect of form drag on ice flow, beyond the scale of valley glaciers.

We have added this recommendation, placing it at Lines 145-148 to conclude our paragraph that generally delivers how we foresee the science being taken forward. Our new sentence reads: “*As an immediate step, the new data now make it possible to run data-informed experiments to develop adequate parameterisations of short-wave form drag on large outlet glaciers and ice streams, and in so doing expand our theoretical knowledge of its effects on ice flow, building upon existing idealised treatments^{43,44}.*” The two new citations make reference to idealised treatments of form drag from valley glaciers and ice streams from Fowler (1979) and Schoof (2002) respectively.

The new sentence here also responds, in part, to the reviewer’s next point.

Line 144: 'Until such independent evaluation of form drag and basal friction is integrated into models, the current generation of ice-sheet models will be hampered in establishing robust projections of ice loss and sea-level rise.'

I agree with hampered, but what is meant by robust? To my knowledge, there are quite a lot of factors that hamper projections of ice loss, and the basal boundary conditions is one amongst many others. That is why ice-sheet modelling actually evolves towards ensembles across parameter uncertainties. Maybe what is needed is theoretical knowledge on how short-wave form drag influences ice flow of large outlet glaciers and ice streams (as opposed to valley glaciers), as I said above.

We agree that using robust in Line 144 is overstating the case and have removed it.

In the new sentence making Lines 145-148 we have now been specific that the new data can be used to improve theoretical knowledge on how short-wave form drag influences ice flow of large outlet glaciers and ice streams.

'such hitherto unimaged high-resolution basal variability is almost certainly affecting rates of ice-stream retreat ': This is also a supposition: for me it is not clear that using old BEDMAP data in modelling part of the paper that the proof of concept is given. The basal variability affects the spatial distribution of basal friction and the basal characteristics, which have an effect on the flow of ice. Whether this influences the rate of ice-stream retreat (in terms of a significant contribution) depends on many other factors as well. Until demonstrated through detailed modelling and comparison, there is no evidence given in the paper.

The reviewer’s quote above is taken from our 18 August rebuttal letter. S/he is fair in stating that we show that the basal [topographic] variability affects the flow of ice, but that we do not necessarily show that this imposes an effect on the ice-stream retreat (a.k.a. ice loss). S/he is also fair in stating (above, third paragraph of Reviewer#2 comments) that the results do show the “potential [of the diverse landscapes] to influence

the rate of ice loss." We therefore think the new title now fairly states the case with which we all (authors and reviewer) agree, i.e.:

"Diverse landscapes beneath Pine Island Glacier influence ice flow and potentially rates of ice loss" (Lines 1-2).

although, as also suggested above, we could entitle the manuscript simply:

"Diverse landscapes beneath Pine Island Glacier influence ice flow"

A novel aspect is definitely that 'hence we can now state with confidence that the basal traction (hence ice flow) of Pine Island Glacier is much more heavily influenced by form drag, i.e. as opposed to basal friction, than has previously been shown to be the case.'

This was also a quote from our response letter, which, thanks to the reviewer's guidance, we have now used explicitly within the paper - Lines 133-135.

Reviewer #3 (Remarks to the Author):

All 3 reviewers were in agreement that this paper presents an interesting new dataset, collected in challenging terrain, using a technically sound approach. The authors have fully addressed the request from reviewers #R1, #R2 & #R3 for more quantitative information on the height, width and roughness of the bed topography (L72-82 and columns 5 to 8 of Table 1), and the request for more attention to errors from #R3 (Table 2, L394-398). There was consensus amongst all 3 reviewers that the dataset is robust, and described well by this paper.

#R2 and #R3 were in agreement that "While the authors do a splendid job on presenting the dataset, the subsequent analysis and consequences for ice sheet modeling has yet to be demonstrated [#R2]." This was a major concern for reviewers #R2 and #R3, because "most of the novelty of this paper hinges on the high resolution bed topography having a direct influence on ice flow, the rate of ice loss, and future sea level rise contribution [#R3]." Both reviewers #R2 and #R3 requested that the authors conduct additional modelling work to "demonstrate this quantitatively instead of making a meager comparison with old data [#R2]."

The authors have not satisfactorily addressed this request, because they have declined to carry out any additional modeling work. The authors justify this decision because they think "basal variability is almost certainly affecting rates of ice-stream retreat", and they would like a future publication to "prove that the various models that have hitherto relied on unrealistically smooth beds still provide adequate bases for projection." The title of this paper states that "Diverse landscapes beneath Pine Island Glacier influence rates of ice loss", however as the authors have admitted in their response above, they have not demonstrated that this 'is' the case.

We accept the criticism of the previous title, as per responses to Reviewer#2 above. We are now careful to state that the diverse landscapes influence ice flow, and to be specific that in doing so they potentially influence ice loss.

While the new Figure 3a enables the reader to more clearly visualise the relationship between basal traction and observed roughness, its not clear to me that there is a statistically significant correlation between the two parameters, and frustratingly statistics about the strength of the correlation that may support this is once again not provided. There is an appearance of more quantitative evidence in the manuscript due to the addition of Table 2, however the numbers provided in columns 1 to 4 are just copy

pasted from the original annotations in Figure 1, therefore don't provide new information to address the concerns of reviewers #R2 and #R3.

We consider that Figure 3 shows the potential relations between basal roughness and basal traction in an appropriately considered way. We are not convinced that there are sufficient data points with sufficient spread across the two parameters to justify placing a quantitative value on the correlation. The other reviewers have not taken issue with this point.

This paper remains a good description of a new dataset, that will be a valuable contribution to a future modelling study. However, it is the modelling study that will represent an important advance of significance to specialists in the field: by informing us whether the existing coarse resolution bedrock topography datasets have affected the modelled rate of ice flow, to the extent that the projections of future sea level rise assumed today are in fact incorrect.

We agree that the follow-on modelling studies, which we too advocate, will certainly provide a significant advance. Nevertheless, we disagree with the implied stance that the current paper is not providing a significant advance in itself. To re-state from our 18 August 2017 rebuttal letter, all three reviewers have already noted the importance of the data and their unprecedented geographical spread at such high resolution beneath glaciated terrain anywhere in the world, let alone a glacier of such high scientific interest.